# Deletion of Mfsd2b impairs thrombotic functions of platelets

Madhuvanthi Chandrakanthan [1,8], Toan Quoc Nguyen[1,8], Zafrul Hasan[1,8], Sneha Muralidharan[2], Thiet Minh Vu[1,7], Aaron Wei Liang Li [2], Uyen Thanh Nha Le[1], Hoa Thi Thuy Ha[1], Sang-Ha Baik[1], Sock Hwee Tan[2], Juat Chin Foo [3], Markus R. Wenk[1,3], Amaury Cazenave-Gassiot [1,3], Federico Torta[1,3], Wei Yi Ong[4], Mark Yan Yee Chan[2] & Long N. Nguyen [1,3,5,6✉]

We recently discovered that Mfsd2b, which is the S1P exporter found in blood cells. Here, we report that Mfsd2b is critical for the release of all S1P species in both resting and activated platelets. We show that resting platelets store S1P in the cytoplasm. After activation, this S1P pool is delivered to the plasma membrane, where Mfsd2b is predominantly localized for export. Employing knockout mice of Mfsd2b, we reveal that platelets contribute a minor amount of plasma S1P. Nevertheless, Mfsd2b deletion in whole body or platelets impairs platelet morphology and functions. In particular, Mfsd2b knockout mice show significantly reduced thrombus formation. We show that loss of Mfsd2b affects intrinsic platelet functions as part of remarkable sphingolipid accumulation. These findings indicate that accumulation of sphingolipids including S1P by deletion of Mfsd2b strongly impairs platelet functions, which suggests that the transporter may be a target for the prevention of thrombotic disorders.

---

[1] Department of Biochemistry, Yong Loo Lin School of Medicine, National University of Singapore, Singapore, Singapore. [2] Department of Medicine, Yong Loo-Lin School of Medicine, National University of Singapore, Singapore, Singapore. [3] Singapore Lipidomics Incubator (SLING), Life Sciences Institute, National University of Singapore, Singapore, Singapore. [4] Department of Anatomy, Yong Loo-Lin School of Medicine, National University of Singapore, Singapore, Singapore. [5] Cardiovascular Disease Research (CVD) Programme, Yong Loo Lin School of Medicine, National University of Singapore, Singapore, Singapore. [6] Immunology Program Research Programme, Life Sciences Institute, National University of Singapore, Singapore, Singapore. [7] Present address: NTT Hi-Tech Institute, Nguyen Tat Thanh University, Ho Chi Minh City, Vietnam. [8] These authors contributed equally: Madhuvanthi Chandrakanthan, Toan Quoc Nguyen, Zafrul Hasan. ✉email: bchnnl@nus.edu.sg

Sphingosine-1-phosphate (S1P) is a lipid mediator that is required for lymphocyte egress from lymphoid organs as well as essential for blood vessel development and functions. S1P exerts its roles by being a ligand for five different S1P receptors (S1P1-5)[1]. In cells, S1P is synthesized by two sphingosine kinases, namely SphK1 and 2[2]. These cellular sources of S1P provide the signal for lymphocyte trafficking, maintenance of blood vessel integrity, and many other biological functions[3]. Endothelial cells (ECs) and blood cells are reported to be the main physiological contributors of S1P in blood[2]. The mechanism by which S1P is released from these cell types was recently unraveled with the cloning of Spns2 and Mfsd2b as the respective S1P transporters in endothelium and blood cells[4,5]. The identification of Spns2 about a decade ago has shed light on the roles of S1P signaling for lymphocyte trafficking[6,7]. Our recent work identified Mfsd2b as the S1P exporter in blood cells. We showed that Mfsd2b contributes the major pool of plasma S1P[5]. Nevertheless, the physiological roles of the Mfsd2b-S1P axis are unclear. Quantitatively, erythrocytes appear to be the main source of plasma S1P by a continuous secretion mechanism. In contrast, platelets are able to store and release S1P upon activation. For example, S1P release from platelets was shown to be responsible for maintenance of high endothelial venules via induction of VE-cadherin[8]. However, it remains elusive whether platelet-derived S1P also contributes to the S1P pool in blood, especially at thrombotic and blood-clotting conditions.

Platelets can synthesize S1P via sphingosine kinase 2 (SphK2)[9]. They take up exogenous sphingosine for S1P synthesis but lack enzymes to hydrolyze S1P[10,11]. Deletion of SphK2 in mice results in loss of S1P synthesis in platelets and reduced platelet biogenesis and functions. Interestingly, S1P release from platelets is inhibited by cyclooxygenases (COX1/2) inhibitors such as aspirin and ibuprofen, implying that this process is dependent on prostaglandin/thromboxane formation[12]. However, whether these COX1/2 inhibitors directly inhibit S1P synthesis or release is uncharacterized. Studies in mice suggest that S1P signaling is involved in platelet biogenesis. Knockout of S1P4 results in delayed platelet production and megakaryogenesis phenotypes, suggesting that S1P-S1P4 receptor signaling is necessary for platelet biogenesis and functions[13]. Platelet biogenesis and production from megakaryocytes were also reported to be reduced in S1P1 knockout mice[14]. However, the signaling roles of S1P from platelets requires further investigations as some parts of the phenotypes were not recapitulated in a recent study[15]. Nevertheless, S1P produced by platelets is essential for several important physiological conditions including anaphylactic shock[16]. Additionally, other recent reports also proposed that there would be more than an S1P transporter in platelets[17]. The cloning of Mfsd2b by our group offered a new approach to study the roles of S1P in platelets[5].

In this study, we report an essential role of Mfsd2b as the S1P exporter in platelets. Surprisingly, we unraveled a homeostatic role of Mfsd2b-S1P pathway in platelets by using the global and platelet-specific knockout of Mfsd2b. We demonstrate that Mfsd2b is required for S1P release in resting and activated platelets and that ablation of Mfsd2b strongly affects the intrinsic functions of platelets that result in reduced thrombosis in mice. Our data support a mechanism in which inhibition of Mfsd2b causes lipotoxicity in platelets and this pathway may represent a novel strategy to reduce the functions of platelets in thrombotic disorders.

## Results

### Mfsd2b is required for S1P release from platelets

Platelets store S1P and release this lipid mediator upon activation. Whether Mfsd2b regulates this process in platelets is unclear. To gain insights into the transport mechanisms of S1P in resting and activated platelets, we activated platelets with thrombin and examined the expression levels of Mfsd2b. We found that expression level of Mfsd2b from resting platelets is similar to that of thrombin-activated platelets (Fig. 1a). As a control, thrombin stimulated release of PF4, a soluble protein stored in alpha-granules, was found to be similar in both wild type (WT) and Mfsd2b knockout (KO) platelets, as shown by a marked reduction in the cell lysate fraction (Fig. 1a). Expression pattern of Mfsd2b was similar to that of Tmem16F, a phosphatidylserine scramblase located in the plasma membrane of platelets (Fig. 1a)[18]. These results show that Mfsd2b localization is associated with platelet membranes.

Next, we used [3-³H]sphingosine to test whether Mfsd2b is responsible for S1P release in resting and thrombin-activated platelets. We found that resting WT platelets released ~34.5% more S1P and 1.8-fold more S1P compared with resting KO cells (Fig. 1b, c). In contrast, thrombin stimulated WT platelets to release 78.8% more S1P and 3.3-fold more S1P compared to KO cells (Fig. 1b, c). As a result, resting WT cells accumulate higher intracellular S1P levels compared with thrombin-activated WT cells (Fig. 1c). These data indicate that resting platelets can store a significant amount of S1P and this S1P pool is released upon activation. Importantly, Mfsd2b KO platelets are unable to release S1P regardless of activation state (Fig. 1b, c). Similar results were also obtained using thin layer chromatography (TLC) analysis (Supplementary Fig. 1a–d). These findings indicate that Mfsd2b is required for S1P export in platelets. To gain further evidence that thrombin-activated WT platelets are unable to store S1P, we performed a time course for S1P release. Indeed, intracellularly synthesized S1P is continuously released into medium and intracellular S1P level is not significantly increased over time in thrombin-activated WT platelets (Fig. 1d, e). These results indicate that activated WT platelets cannot store S1P. In contrast, activated KO platelets are unable to release S1P, thus accumulating it over time. The inability to release S1P is not due to a failed release of granular contents as Mfsd2b KO platelets were able to release ATP, PF4, and serotonin in similar manners to that of WT platelets after thrombin activation (Supplementary Fig. 2a–d). Together, our results show that Mfsd2b is required for S1P release from resting and activated platelets.

### Mfsd2b is predominantly expressed in the plasma membrane, whereas S1P is mainly stored in the cytoplasm in resting platelets

The ability of resting platelets to store a pool of S1P and release it after activation remains elusive. We argue that if S1P which is integrated in lipid membranes and Mfsd2b which is a membrane protein are co-localized, platelets cannot store S1P in a similar fashion to erythrocytes. To test whether Mfsd2b is localized in the plasma membrane of resting platelets, we performed the cell-surface biotinylation assay. We detected Mfsd2b protein in the plasma membrane (neutravidin bound) fractions in both resting and activated platelets (Fig. 2a, b). In comparison with resting platelets validated by the presence of intracellular PF4, similar expression levels of Mfsd2b were found in the plasma membrane of activated platelets that are characterized by the absence of PF4 (Fig. 2a, b). Mfsd2b expression patterns in resting platelets were similar to that of integrin b3, a plasma membrane maker (Fig. 2a, b). When normalized to beta-actin as an intracellular marker, the expression levels of Mfsd2b in the plasma membrane from resting and activated platelets were significantly higher than its levels in the cytosolic fractions (Fig. 2c, d). In addition, we performed subcellular fractionation of cytosolic and membrane fractions from resting WT platelets using freeze-thaw cycles[19]. Our results showed that Mfsd2b

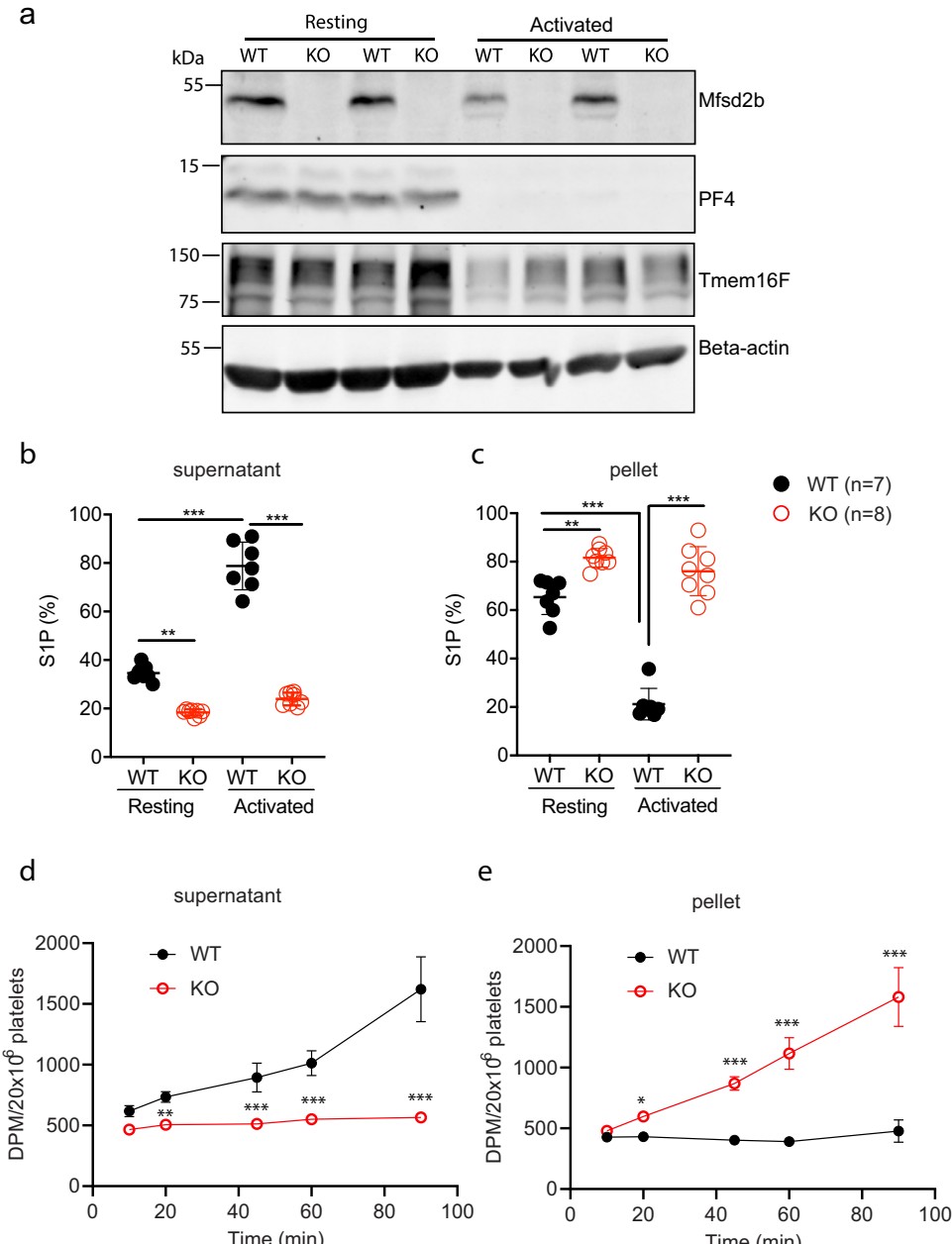

**Fig. 1 Mfsd2b is expressed in resting and activated platelets and is required for S1P release. a** Mfsd2b is expressed in resting and thrombin-activated platelets with comparable levels. Western blot analysis of Mfsd2b expression in resting and thrombin-activated platelets from WT and global knockout of Mfsd2b (KO). Mfsd2b expression is retained in thrombin-activated platelets at levels similar to resting cells. Note that PF4 is a soluble protein and it is strongly released from alpha-granules after activation with thrombin in both WT and KO platelets. WT wild-type, KO knockout. Tmem16F is a membrane protein of platelets. Experiments were repeated twice ($n = 2$). **b**, **c** Resting and thrombin-activated platelets export S1P in Mfsd2b-dependent manners. Resting and thrombin-activated platelets from WT and KO mice were incubated with [3-$^3$H]-sphingosine in Tyrode H buffer containing 0.5% BSA for 1 h at 37 °C. The cell pellets and supernatants were collected for S1P determination. Scintillation quantification of S1P levels from supernatants and cell pellets expressed as percentage. Data are mean and SD. Experiments were repeated twice ($n = 7$–8). **$P < 0.01$, ***$P < 0.001$; ns not significant. $P$ values were calculated using one-way ANOVA. **d**, **e** Time course transport assays for thrombin-activated platelets (platelets were activated before transport assays). Activated WT platelets readily release S1P in an Mfsd2b-dependent manner. Note that activated WT platelet cannot store S1P. Data are mean and SD. Experiments were repeated twice ($n = 4$). *$P < 0.05$, **$P < 0.01$, ***$P < 0.001$. $P$ values were calculated using one-way ANOVA. DPM disintegrations per minute.

expression levels were predominantly found in the membrane fractions (Fig. 2e, f). Consistent with the biotinylation results above, Mfsd2b localization was coincided with the localization of intergrin b3 (Fig. 2g). Together, these results provide strong evidence that Mfsd2b is mainly localized in the plasma membrane of both resting and activated platelets.

Additionally, we performed subcellular fractionation of cytosolic and membrane fractions from resting WT platelets in which the cells were preloaded with [3H]-S1P. S1P levels from cytosolic fractions were significantly greater than that from membrane fractions (Fig. 2h). Furthermore, we detected significantly higher levels of endogenous S1P harvested from cytosolic fractions from

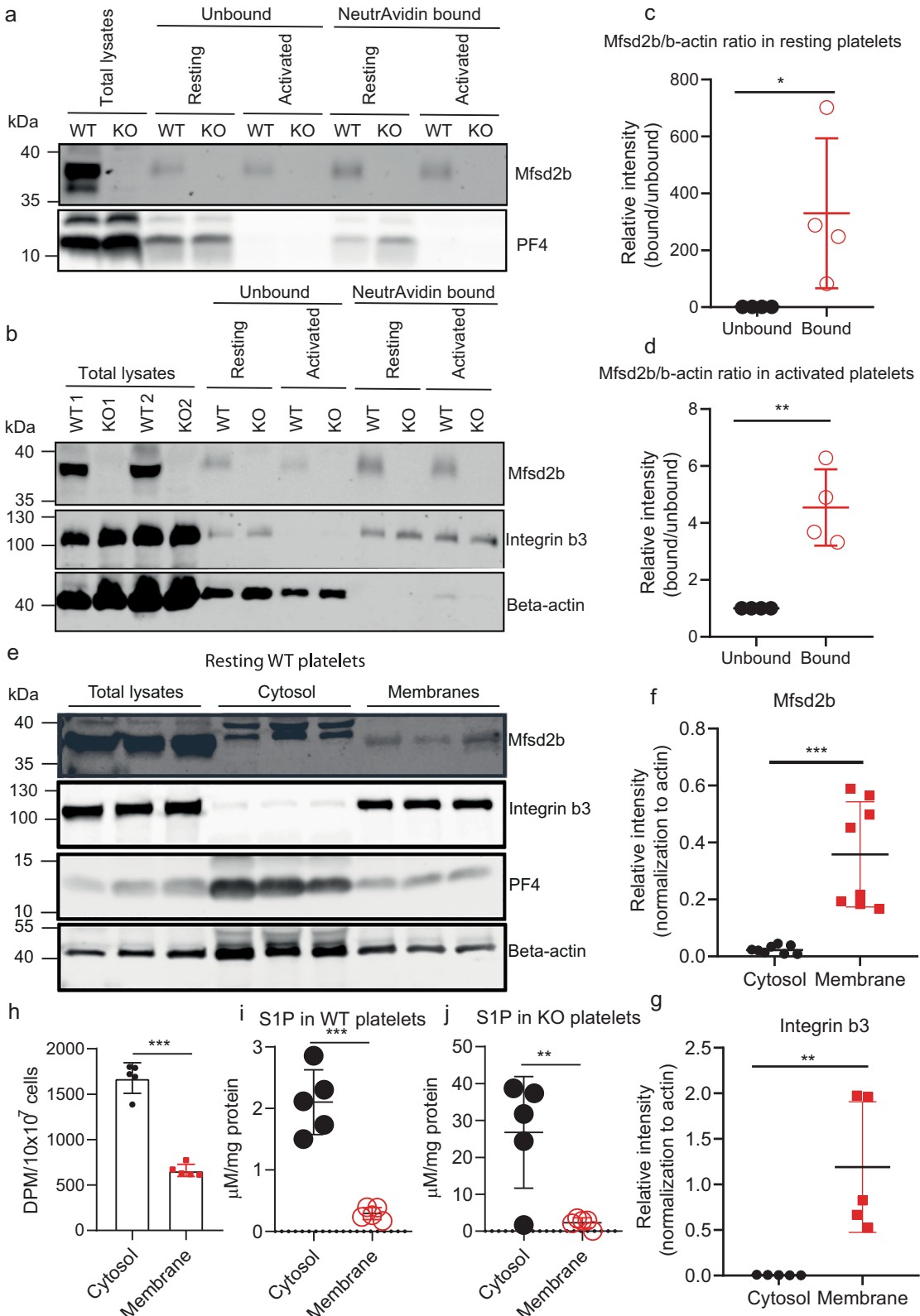

WT and Mfsd2bf/fPF4 knockout platelets compared with that from membrane fractions (Fig. 2i, j). These results show that in resting platelets S1P is predominantly localized in the cytoplasm. Together, our results show that Mfsd2b is mainly expressed in the plasma membrane, whereas most of the S1P content is found in the cytoplasm in resting platelets. These results likely explain the ability of storing S1P in resting platelets.

**Mfsd2b is responsible for export of different S1P species.** Platelets contain different S1P species, which are yet to be determined as Mfsd2b ligands. Additionally, some other proteins have also been associated as S1P transporters in platelets[17,20,21]. To determine whether Mfsd2b exports a specific- or all S1P species in a promiscuous manner, we performed mass spectrometry analysis for detectable S1P species in resting and

**Fig. 2 Mfsd2b is mainly expressed in the plasma membrane of platelets. a, b** biotinylation experiments of resting and thrombin-activated platelets WT and Mfsd2bf/f PF4 KO platelets to detect the expression of Mfsd2b in the plasma membrane. Biotin-bound (NeutrAvidin) and unbound fractions were collected from resting and thrombin-activated platelets for detection of Mfsd2b, integrin b3, a plasma membrane marker, and beta-actin, a cytosolic protein. Experiments were repeated at least three times. WT wild-type, KO knockout. **c, d** Quantification of Mfsd2b protein band from biotin-bound and unbound fractions harvested from resting and activated platelets. The intensity of Mfsd2b bands was normalized to beta-actin bands. Mfsd2b is significantly expressed in the plasma membrane of resting and activated platelets. Data are mean and SD. Each dot represents one mouse ($n = 4$). *$P < 0.05$, **$P < 0.01$, ***$P < 0.001$. $P$ values were calculated by two-tailed unpaired $t$-test. **e** Subcellular fractionation of membrane and cytosolic proteins from resting WT platelets. Proteins from the membrane and cytosolic fractions were probed with Mfsd2b, integrin b3, PF4, and beta-actin antibodies. Similar to integrin b3, expression of Mfsd2b is significantly higher in the membrane fractions. Experiments were repeated three times. **f, g** Quantification of Mfsd2b and integrin b3 from membrane and cytosolic fractions, respectively. Data are mean and SD. Each dot represents one mouse ($n = 8$ in **f** and $n = 5$ in **g**). **$P < 0.01$, ***$P < 0.001$, two-tailed unpaired $t$-test is used. **h** Radioactive S1P in cytosolic and membrane fractions from resting WT platelets. Data are mean and SD. Each dot represents one mouse ($n = 5$). ***$P < 0.001$, two-tailed unpaired $t$-test. **i, j** mass spectrometry analysis of endogenous S1P in cytosolic and membrane fractions from resting platelets from WT and Mfsd2bf/fPF4 mice. Total S1P levels (sum of all measured S1P species) were used. Data are mean and SD. Each dot represents one mouse ($n = 5$). **$P < 0.01$, ***$P < 0.001$, two-tailed unpaired $t$-test. The quantification bands (in **c, d, f, g**) were derived from the same experiments and the blots were processed in parallel. DPM disintegrations per minute.

thrombin-activated platelets from WT and Mfsd2b KO mice. Lipidomic analysis of S1P species including S1P(d18:0), S1P (d18:1), and S1P (d18:2) in thrombin-activated WT and Mfsd2b KO platelets showed that after stimulation, WT platelets released these endogenous S1P species from the cells (Fig. 3a). We estimated that ~80% of every S1P species is equally released after activation, suggesting that Mfsd2b transports different S1P species in a promiscuous manner. In contrast, these S1P species are all retained in resting and thrombin-activated KO platelets with similar levels, indicating that Mfsd2b is responsible for exporting these S1P species in platelets (Fig. 3a, b). Our results indicate that Mfsd2b is the promiscuous transporter for S1P species in platelets.

Platelets contain a variety of other lipids, which are known to have significant roles in their functions[22,23]. Consistently with Mfsd2b as a specific transporter for S1P species, lipidomic analysis data showed that the levels of phospholipids such as phosphatidylcholines (PC), and phosphatidylethanolamines (PE) from resting and activated WT and Mfsd2b KO platelets were unchanged (Supplementary Fig. 3). Furthermore, we found that small levels of sphingosines and ceramides, but not sphingomyelins (SM), which are the possible precursors for S1P synthesis were accumulated in Mfsd2b KO platelets (Fig. 3c, d). Platelets are known to assimilate exogenous sphingosines for S1P synthesis via SphK2[9,10,16]. We also noted that sphingosine uptake was sufficient in resting or activated KO platelets in our transport assays. As the de novo synthesis of sphingolipids is not found in platelets[10], it is unlikely that hydrolysis of SM increases the levels of ceramides and sphingosines in KO platelets. The increases in ceramides and sphingosines suggests that the remarkably high amount of S1P accumulation is converted back to these lipids (Fig. 3c, d), perhaps due to the presence of S1P phosphatases[11]. Together, our data indicate that Mfsd2b specifically exports all S1P species and that loss of Mfsd2b specifically affects sphingolipid levels in platelets.

**Platelets are not a major source of plasma S1P under normal condition.** We previously showed that Mfsd2b is expressed in erythrocytes and platelets. The whole-body knockout mice of Mfsd2b had ~50% S1P reduction in plasma[5]. It is uncharacterized how much S1P is contributed from platelets compared with that from erythrocytes. To determine whether platelets contribute to plasma S1P generation in the circulation, we measured S1P levels in plasma and platelets from platelets specific knockout of Mfsd2b (Mfsd2bf/fPF4) and erythrocytes specific knockout of Mfsd2b (Mfsd2bf/fEpoR). First, we showed that Mfsd2b expression is successfully abolished in the respective knockout mice (Fig. 4a, b). Secondly, deletion of Mfsd2b in platelets also inhibited S1P export functions in washed platelets isolated from global KO and

Mfsd2bf/fPF4 mice, whereas erythrocytes from Mfsd2bf/fPF4 retained a normal S1P transport activity (Fig. 4c, d). Then, we performed lipidomic analysis of plasma and platelets isolated from the mice. Our data showed that similar to the global deletion of Mfsd2b, deletion of Mfsd2b in erythrocytes significantly reduces plasma S1P compared with WT or controls (Fig. 4e). There is a similar reduction in plasma S1P level from whole-body Mfsd2b knockout and Mfsd2bf/fEpoR, indicating that erythrocytes are indeed the major sources of plasma S1P (Fig. 4e). Deletion of Mfsd2b in platelets did not significantly reduce plasma S1P level (Fig. 4e), whereas there were increased levels of S1P accumulation in the platelets from KO or Mfsd2bf/fPF4 mice (Fig. 4f). These results indicate that platelets are not a major source of plasma S1P under normal condition. Although deletion of Mfsd2b resulted in S1P accumulation in platelets, we reason that the S1P pool from platelets might be insufficient to increase blood S1P, where erythrocytes constitutes the major cell types. Together, our data imply that platelets only provide a minimal amount of S1P to plasma.

**Mfsd2b knockout platelets exhibit reduced aggregation.** In activation assays using whole blood, we found that whole blood from Mfsd2b KO mice has reduced aggregation upon stimulation with ADP (Fig. 5a). Similarly, blood from Mfsd2b KO mice has reduced aggregation upon activation with the calcium ionophore A23187 (Fig. 5a). Protease-activated receptor-4 peptide (PAR-4) activation or aggregation induced by collagen is also reduced in KO platelets (Fig. 5a). In agreement with the whole blood aggregation assays, washed platelets isolated from global Mfsd2b KO mice have reduced activation using ADP and A23187 (Fig. 5b). The reduced aggregation is not due to the altered expression of adhesion molecules CD42a, CD42b, CD42c, CD42d, GPVI, and CD61 (Supplementary Fig. 4). Notably, there is a significant reduction in the expression of αIIbβ3 (JON/A) and P-selectin on the surface of activated platelets from KO mice compared with that of WT mice after activation with 1 μM A23187 (Fig. 5c, d). These results suggest that the accumulation of sphingolipids in Mfsd2b KO platelets may affect the exposure of adhesion molecules upon activation.

**Mfsd2b knockout platelets exhibit abnormal morphologies.** The impaired functions of Mfsd2b knockout platelets are unexpected given the recent report of dispensable roles of S1P signaling in platelets[15]. However, we noted that Mfsd2b KO platelets had altered morphology in complete blood count analysis (Supplementary Fig. 5a, b). Using electron microscopy, we observed that resting Mfsd2b KO platelets exhibited multiple lesions and pores in the plasma membrane by SEM (Fig. 6a, b; arrowheads).

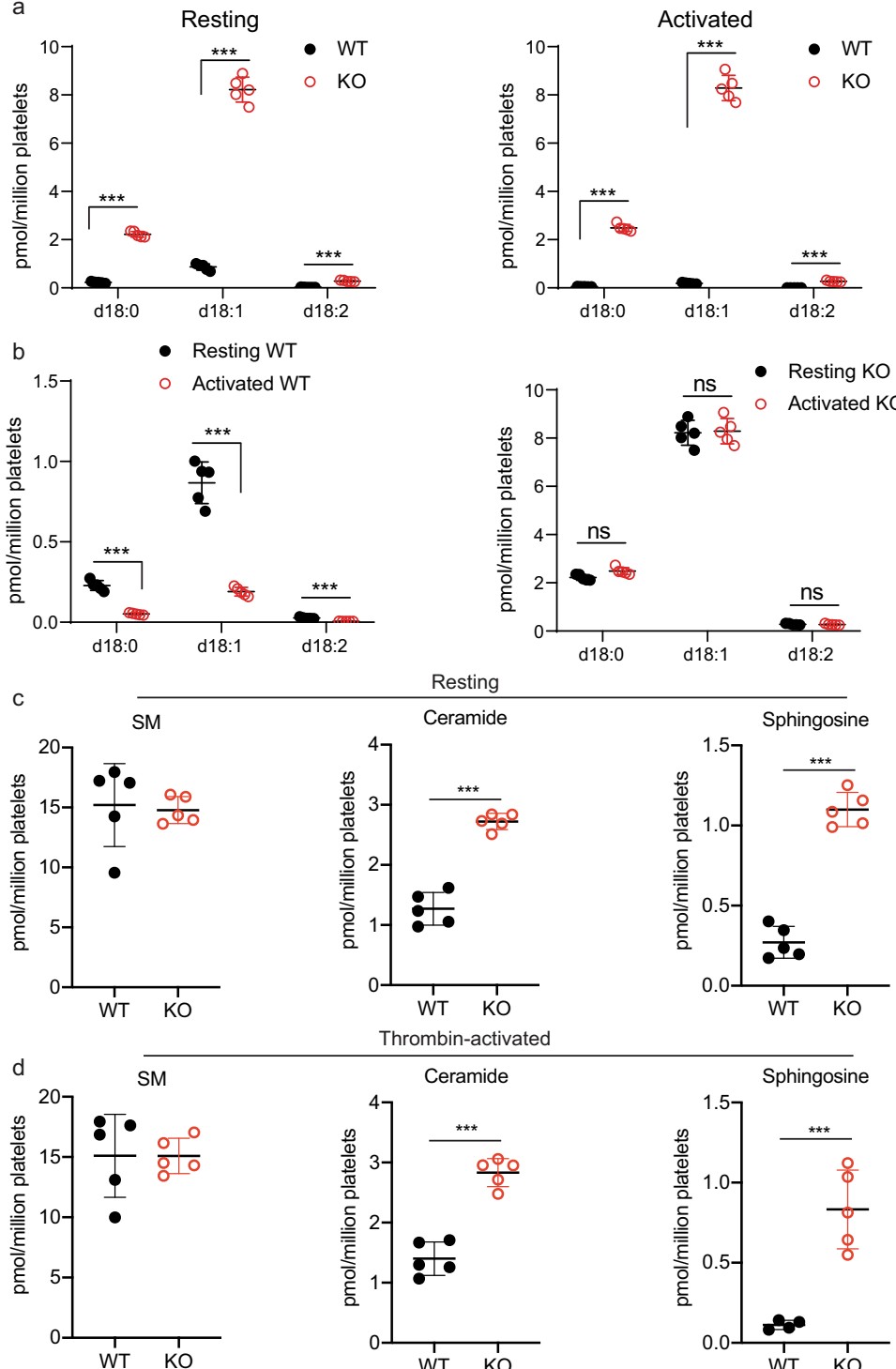

**Fig. 3 Mfsd2b is likely a promiscuous S1P transporter. a** Levels of individual S1P species from resting and thrombin-activated WT and Mfsd2b KO platelets. These are major S1P species found in platelets. Data are mean and SD. Each dot represents one mouse ($n = 5$). ***$P < 0.001$. One-way ANOVA was used. **b** comparison of the levels of each S1P species measured from resting and thrombin-activated WT and KO platelets, respectively. There was a reduction in S1P levels in WT platelets at activated conditions (right panel). Thrombin-activation did not reduce S1P levels in Mfsd2b KO platelets (left panel). Data are mean and SD. Each dot represents on mouse ($n = 5$). ***$P < 0.001$; ns not significant. One-way ANOVA was used. WT wild type, KO knockout. **c**, **d** Levels of sphingomyelins (SM), ceramides, and sphingosines found in WT and Mfsd2b KO platelets before and after thrombin activation, respectively. Levels are the sum of all measured molecular species in each lipid class. The levels of ceramides and sphingosines were slightly elevated in KO platelets. However, note that the degree of S1P accumulation is much greater compared with ceramides and sphingosines. Data are mean and SD. Each dot represents one mouse ($n = 5$). ***$P < 0.001$, two-tailed unpaired $t$-test was used. Individual lipid species can be found in the supplementary data 1. SM sphingomyelin.

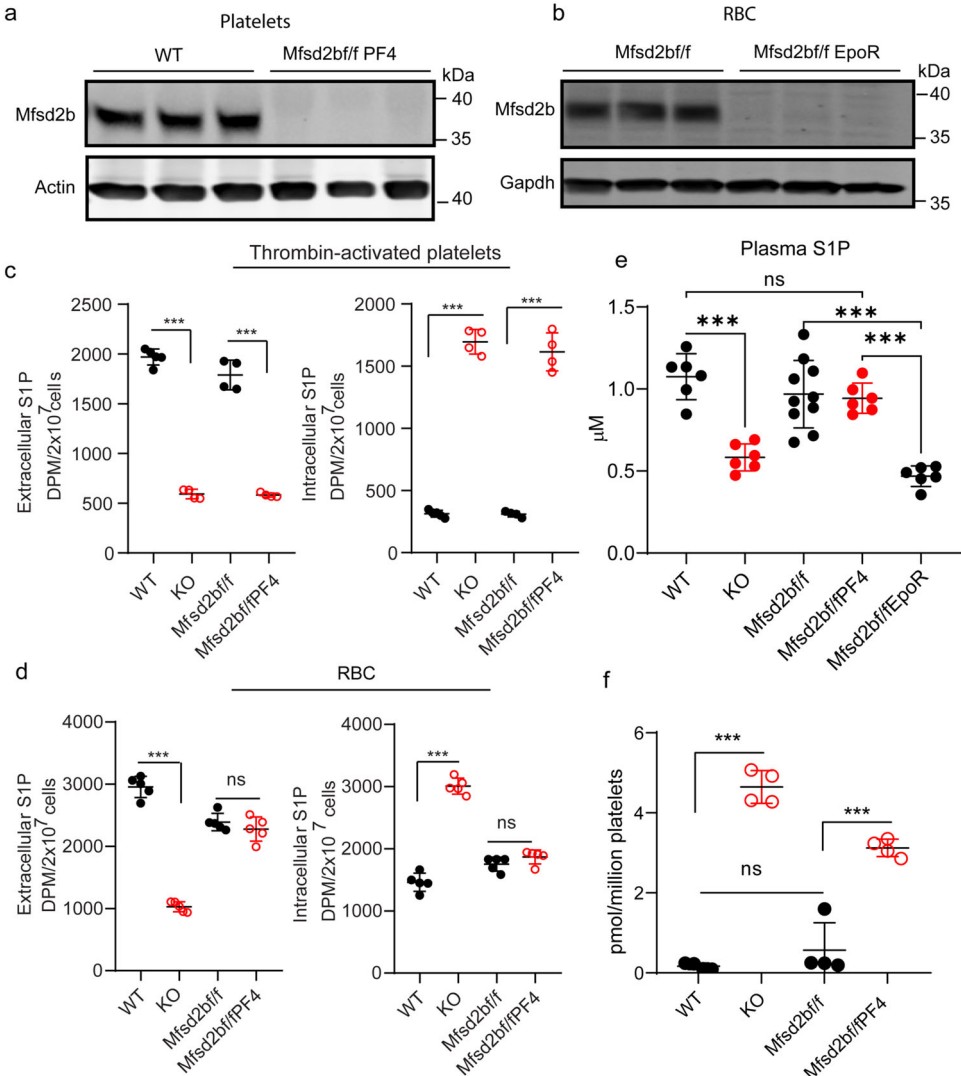

**Fig. 4 Platelets play a dispensable role for plasma S1P. a, b** Western blot analysis of Mfsd2b expression in platelets and erythrocytes from WT, Mfsd2bf/fPF4, and Mfsd2bf/fEpoR mice. Experiments were performed at least twice with n = 3. WT wild-type, KO knockout, RBC red blood cells. **c** Extracellular and intracellular S1P levels from transport assays with thrombin-activated platelets from the indicated mice to evaluate the ability of S1P export. Similar to platelets from global knockout of Mfsd2b, platelets from PF4-Cre specific knockout of Mfsd2b (Mfsd2bf/fPF4) had a reduced S1P transport activity comparable to platelets isolated from global KO and control (Mfsd2bf/f). Data are mean and SD. Experiments were performed at least twice with n = 4. ***P < 0.001; ns not significant. One-way ANOVA was used. **d** Extracellular and intracellular S1P levels from transport assays of isolated erythrocytes to evaluate the ability of S1P export. Erythrocytes isolated from Mfsd2bf/fPF4 had normal S1P transport activity, whereas S1P export activity is reduced in global Mfsd2b KO erythrocytes. Data are mean and SD. Experiments were performed at least twice with n = 4. ***P < 0.001; ns not significant. One-way ANOVA was used. **e** Total plasma S1P levels from WT (n = 6), global Mfsd2b KO (n = 6), Mfsd2bf/fPF4 (n = 6), Mfsd2bf/fEpoR (n = 6) knockout mice, and Mfsd2bf/f mice (n = 10). Data are mean and SD. ***P < 0.001. ns not significant. One-way ANOVA was used. **f** S1P levels in platelets isolated from the indicated genotypes. Data are mean and SD. Each dot represents one animal (n = 4). ***P < 0.001; ns not significant. One-way ANOVA was used. DPM disintegrations per minute.

Furthermore, transmission electron imaging revealed that KO platelets had increased membrane blebbing (Fig. 6c, d; arrows). We also observed multiple small open canalicular (OCS) systems in KO platelets (Fig. 6d; arrowheads). Deletion of Mfsd2b in erythrocytes resulted in increased membrane invagination, which causes stomatocytes[5]. Additionally, deletion of SphK2 does not result in morphological changes in platelets[9]. Thus, we anticipated that the remarkable increases in sphingolipids, especially S1P, in Mfsd2b KO platelets could cause the observed morphological phenotypes. Thus, we investigated whether the defects in membrane structures of KO platelets affect the spreading phenotypes. Indeed, we found that activated KO platelets exhibit significantly reduced filopodia phenotype but showed increased

spreading lamellipodia compared with that of WT platelets (Fig. 6e, f and Supplementary Fig. 6). These microscopic observations suggest that the accumulated sphingolipids are especially detrimental to platelet morphology and possibly functions.

**Mfsd2b knockout mice show reduced venous thrombosis.** Next, we tested the thrombus formation in vivo. In tail vein bleeding assay, although there is a slightly increased bleeding time in global Mfsd2b KO mice, it is statistically not different in comparison with WT mice (Supplementary Fig. 7). To gain more insights into the effects of Mfsd2b deletion in platelets, we examined the thrombotic capacity of KO platelets in a deep venous thrombosis (DVT) model, by performing a stenosis of the inferior vena cava

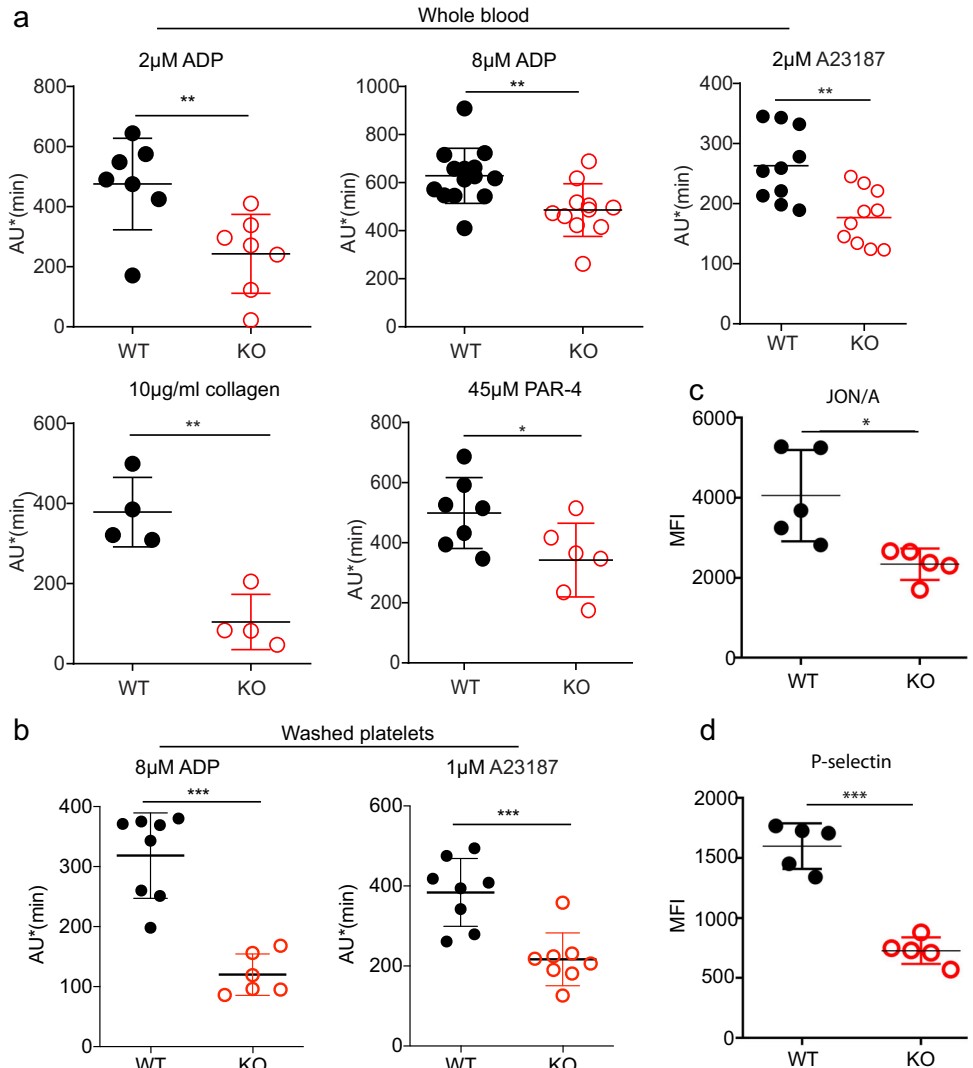

**Fig. 5 Mfsd2b KO platelets exhibited reduced aggregation. a** aggregation assays of whole blood isolated from WT and Mfsd2b KO mice with indicated agonists. Whole blood samples from WT and KO mice were activated with indicated concentrations of agonists. Aggregation ability was measured by an aggregometry analyzer and shown as area under the aggregation curve (AU). Data are mean and SD. Each dot represents one mouse ($n = 7$ for 2 μM ADP activation; $n = 14$ for WT and $n = 11$ for KO for 8 μM ADP activation; $n = 10$ for 2 μM A23187 activation; $n = 4$ for 10 μg/ml collagen activation; $n = 7$ for WT and $n = 6$ or KO for 45 μM PAR-4 activation experiments). **$P < 0.01$, two-tailed unpaired $t$-test was used. ADP adenosine diphosphate, A23187 calcium ionophore, PAR-4 protease-activated receptor-4, JON/A platelet glycoprotein GPIIb/IIIa. **b** aggregation assays of washed platelets isolated from WT and KO mice with indicated agonists. Washed Mfsd2b knockout platelets exhibited significantly impaired aggregation compared with WT controls as observed in the whole blood samples. Data are mean and SD. Each dot represents one mouse ($n = 8$ for WT and $n = 6$ for KO for 8 μM ADP activation; $n = 8$ for 1 μM A23187 activation experiments). ***$P < 0.001$, two-tailed unpaired $t$-test was used. AU area under the aggregation curve. **c**, **d** Mfsd2b knockout platelets had significantly reduced expression of JON/A and P-selectin to the cell surface after activation with 1 μM A23187. Surface expression of JON/A and P-selectin was analyzed by flow cytometry. See supplementary fig. 10 for gating strategy. Data are mean and SD. Each dot represents one mouse ($n = 5$). *$P < 0.05$, ***$P < 0.001$. Two-tailed unpaired $t$-test was used. MFI mean fluorescence intensity.

(IVC) vessel as described previously[24]. We found that global Mfsd2b KO mice show significantly reduced venous thrombosis with reduced size and weight of thrombi (Fig. 7a, b). Notably, blood clots were already observed in WT mice after 6 h of IVC stenosis and the IVC lumen from WT mice was filled up with thrombi 48 h post-stenosis (Fig. 7c, d). Importantly, we observed that IVC isolated from KO mice lacked visible blood clots in IVC after 48 h of stenosis (Fig. 7d). Interestingly, we observed the attachment of white pulps to IVC in thrombi from WT mice (Fig. 7c, arrowhead, and Supplementary Fig. 8a, b). As a possible sign for thrombus formation, we found that the number of circulating platelets from WT mice but not in the KO mice are significantly reduced, implicating that WT platelets are recruited and trapped at the thrombotic sites (Fig. 7e). Number of platelets

is not reduced in blood of Mfsd2b KO mice after IVC stenosis, suggesting that they fail to aggregate at the stenosis sites. Likely, a direct adhesion of platelets to thrombotic sites is critical for thrombosis in DVT as also reported in other models[25,26]. To gain insight whether plasma S1P is altered in thrombotic conditions, we performed S1P measurement from mice that are undergone 48 h IVC stenosis. Interestingly, plasma S1P levels in these thrombotic WT mice are unaltered compared with that of untreated WT controls (Fig. 7f). Thus, unlike anaphylactic shock, systemic S1P is apparently not required for inducing venous thrombosis. Again, these findings indicate that platelets do not contribute a significant amount of circulating S1P. Together, our data show that lack of Mfsd2b prevents platelets from inducing venous thrombosis.

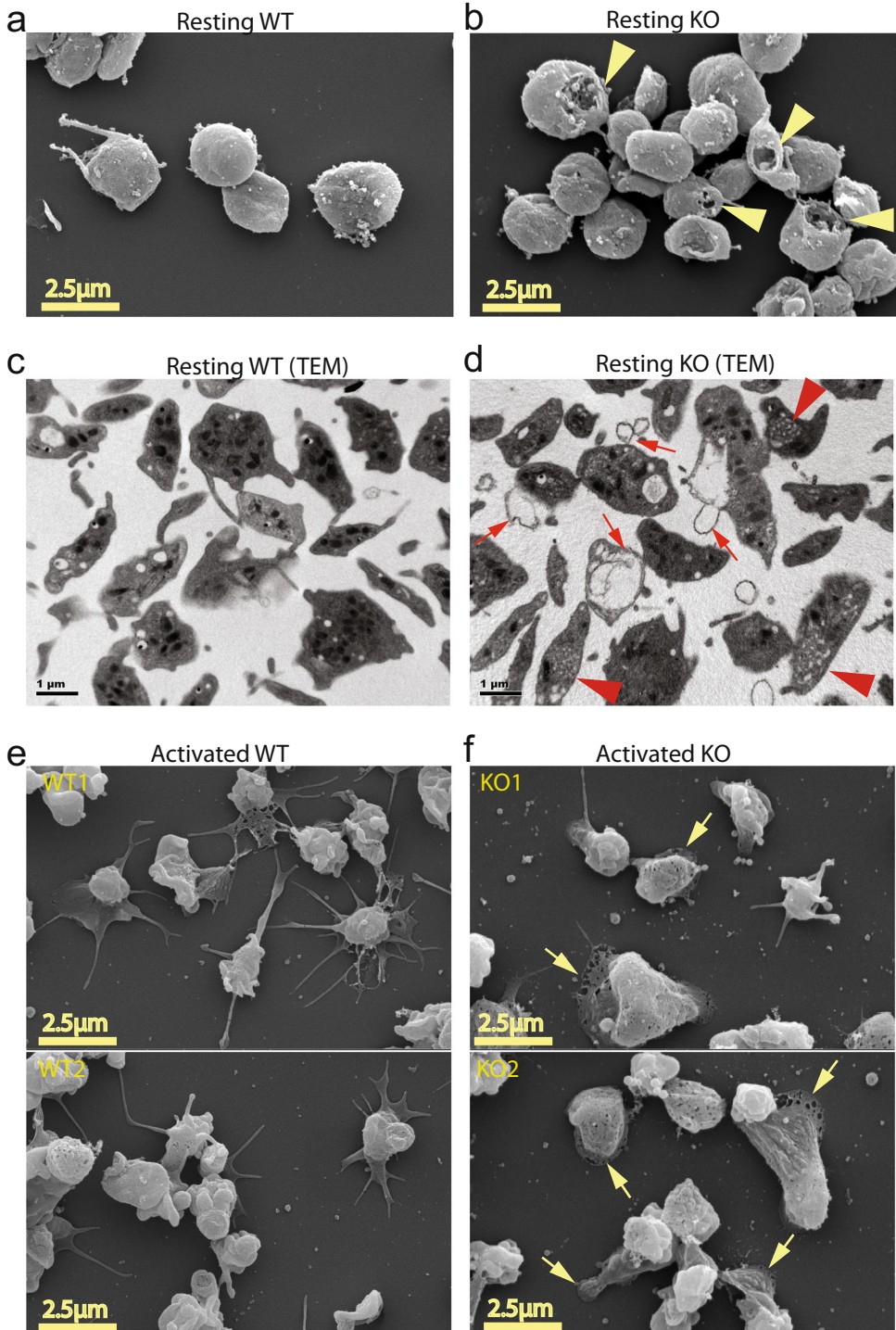

**Fig. 6 Mfsd2b knockout platelets had abnormal morphology. a**, **b** Representative images from scanning electron microscopic (SEM) of resting WT and Mfsd2b KO platelets. Note that KO platelets exhibited membrane damages (shown in yellow arrowheads). WT wild-type, KO knockout**. c**, **d** Representative images from transmission electron microscopic (TEM) of resting WT and KO platelets. Note that KO platelets exhibited increased membrane protrusion and blebbing, likely due to defects in the open canalicular systems (shown in red arrows). There were multiple small open canalicular systems (OCS) observed in KO cells (red arrowheads). **e**, **f** Representative images from scanning electron microscopic (SEM) of calcium ionophore A23187 (1 μM) activated WT and KO platelets. Platelets were activated for 5 min on coverlips and fixed. Note that activated KO platelets had reduced spreading and increased lamellipodia (shown in yellow arrows). In **a**–**d**, representative images from 3 WT and 3 KO mice. In **e**, **f** upper and lower panels are images from platelets isolated from two different animals ($n = 2$).

**Intrinsic defects in Mfsd2b knockout platelets are possibly linked to the reduced platelet functions**. Inflammation can promote DVT[27]. Thus, we examined whether S1P signaling is required to induce vascular inflammation. S1P2 is known to have a critical role in inducing vascular inflammation[28]. We focused on examining whether S1P2 is involved in DVT. However, DVT formation in S1P2 knockout mice is not different from WT controls (Fig. 8a). Furthermore, inhibition of S1P2 with JTE-013,

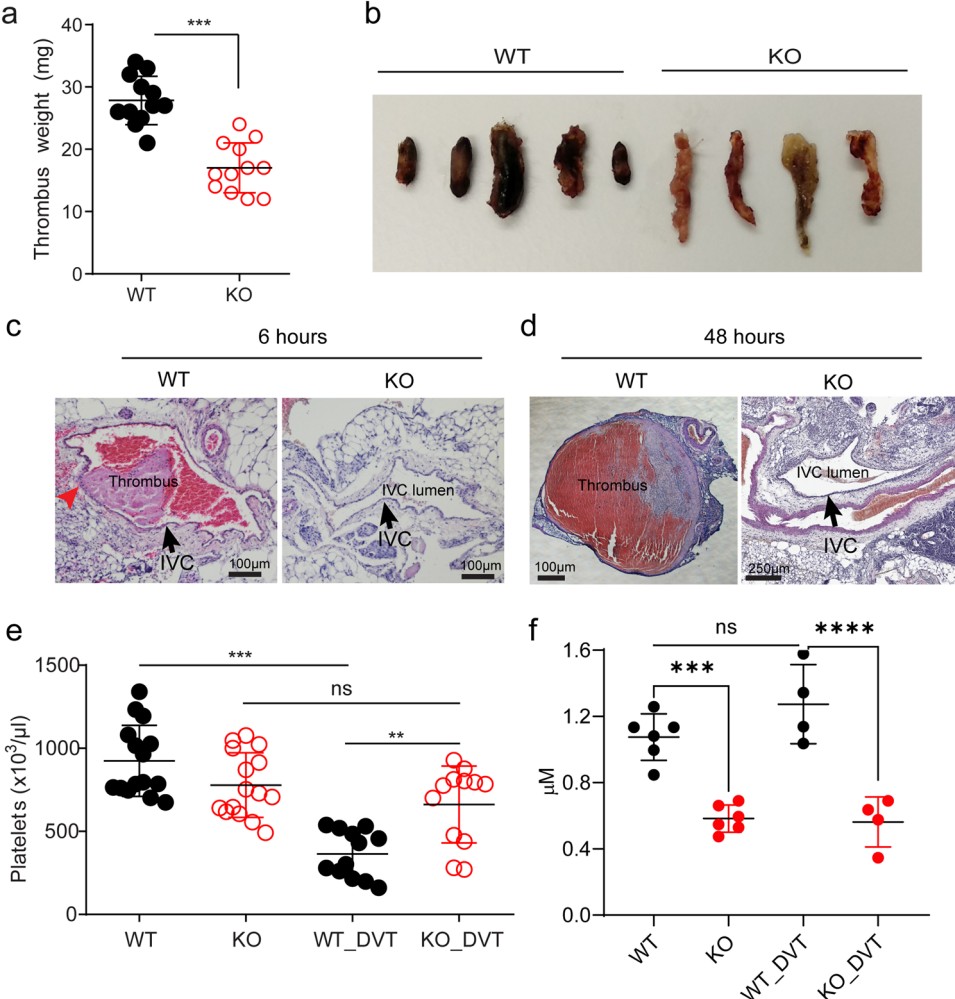

**Fig. 7 Mfsd2b knockout mice show reduced deep vein thrombosis. a**, **b** Global Mfsd2b knockout mice had reduced venous thrombosis in IVC stenosis model. Weights (**a**) and representative images (**b**) of thrombi collected 48 h post stenosis of IVC from WT and KO, respectively. Data are mean and SD. Each dot represents one animal ($n = 12$). ***$P < 0.001$. Two-tailed unpaired $t$-test was used. WT wild-type, KO knockout. **c**, **d** Representative histology of inferior vena cava (IVC) after 6 and 48 h of stenosis in WT and Mfsd2b KO mice. Experiments were performed at least twice ($n = 2$–3). Arrows indicate IVC. **e** Thrombosis resulted in reduced platelets count in blood of WT mice. Data are mean and SD. Each dot represents one animal ($n = 15$ for WT and KO, $n = 12$ for WT-DVT and KO-DVT). **$P < 0.01$, ***$P < 0.001$; ns not significant. One-way ANOVA was used. **f** Venous thrombosis did not induce the loss of total plasma S1P. Plasma S1P from indicated mice was collected before and 48 h after DVT. Note that the analysis of S1P for plasma collected from DVT experiments was acquired at the same time with the data shown in Fig. 4e. Thus, data for S1P levels from WT and global Mfsd2b KO were used for comparison. Data are mean and SD. Each dot represents one animal ($n = 4$–6). ***$P < 0.001$; ****$P < 0.0001$; ns not significant. One-way ANOVA was used. DVT deep vein thrombosis.

a specific antagonist of S1P2 receptor, does not reduce DVT in wild-type mice (Fig. 8b). Thus, it is unlikely that S1P signaling at least via S1P2 is required to induce vascular inflammation in DVT. To gain insights whether Mfsd2b KO platelets have intrinsic impairment of thrombotic functions, we performed platelet transfusion. Transfusion of WT platelets rescued the thrombotic impairment seen in KO mice, whereas transfusion of KO platelets to KO mice did not increase the reduced thrombosis (Fig. 8c). Global deletion of Mfsd2b results in reduction of S1P plasma and concomitant accumulation in erythrocytes. This may contribute to the reduced thrombosis in Mfsd2b KO mice. To demonstrate that Mfsd2b in platelets, but not in erythrocytes is responsible for the observed phenotypes, we assessed DVT formation in erythrocyte-specific knockout of Mfsd2b (Mfsd2bf/fEpoR). Our results showed that thrombus formation in Mfsd2bf/fEpoR mice was similar to that of Mfsd2bf/f controls (Fig. 8d). These results show that KO platelets exhibit intrinsic defects in induction of thrombosis

Platelets adhere to vascular sites via a possible interaction with podoplanin, a surface marker strongly induced in inflammation[29]. Inhibition of platelet adhesion to vascular wall via targeting platelet-derived Clec-2 or vascular-derived podoplanin has been shown to affect DVT in mice[26]. To gain insight into the reduced DVT in Mfsd2b KO mice, we examined expression of podoplanin at the thrombotic sites. We showed that the basal expression of podoplanin is highly induced in IVC isolated from thrombotic WT mice, suggesting that this receptor is involved in thrombotic events (Fig. 8e). Thus, we examined whether expression of podoplanin is reduced in IVC isolated from KO mice. Indeed, our results show that the expression level of podoplanin in IVC isolated from KO mice is significantly reduced compared with that of WT mice (Fig. 8e, f). The interaction of KO platelets from Mfsd2b KO mice with leukocytes was significantly reduced compared with that of WT platelets (Fig. 8g). These results further indicate that deletion of Mfsd2b affects platelets adhesion capacity during thrombosis.

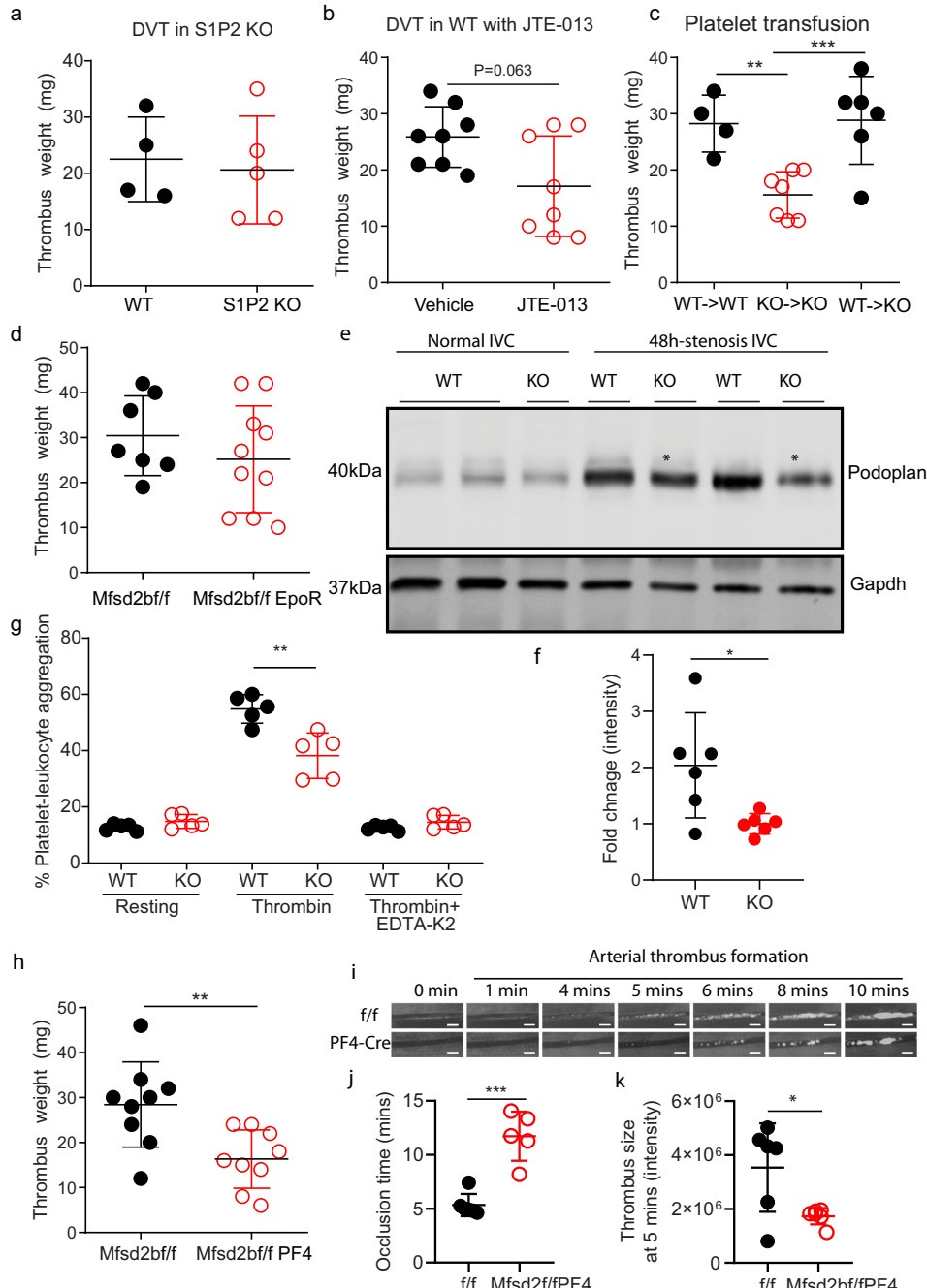

**Fig. 8 Deletion of Mfsd2b in platelets reduces thrombosis. a** S1P2 is dispensable for thrombosis. Weights of thrombi collected from WT and S1P2 knockout mice. Each dot represents one animal ($n = 4$–$5$). **b** JTE-013, a S1P2 antagonist, also failed to inhibit DVT in mice. Weights of thrombi collected from WT mice ($n = 8$) with and without treatment of 30 mg/kg JTE-013. **c** Platelet transfusion. Transfusion of WT platelet to WT or KO mice, respectively and vice versa. The indicated recipient mice were assessed for DVT. Transfusion of KO platelets failed to increase DVT formation in KO mice, whereas transfusion of WT platelets to KO mice reversed the reduced thrombosis in KO mice. Each dot represents one animal ($n = 4$ for WT > WT; $n = 7$ for KO > KO; $n = 6$ for WT > KO). **$P < 0.01$, ***$P < 0.001$. One-way ANOVA was used. **d** Deletion of Mfsd2b in erythrocytes using EpoR-cre did not reduce thrombosis. Each dot represents one animal ($n = 7$–$10$). **e, f** Expression of podoplanin in untreated and stenosis IVC from WT and KO mice. In stenosis IVC, thrombi were removed from IVC for western blot analysis. Each dot represents one mouse ($n = 6$). *$P < 0.05$. Two-tailed unpaired $t$-test was used. **g** Flow cytometry assays of platelets-leukocyte aggregate assays showed that activated Mfsd2b KO platelets exhibited reduced aggregation with leukocytes in comparison with activated WT platelets. See supplementary fig. 10 for gating strategy. Each dot represents one mouse ($n = 5$). **$P < 0.01$. One-way ANOVA was used. **h** Deletion of Mfsd2b in platelets using PF4-cre reduced thrombosis. In the DVT experiments, thrombus weight after 48 h of stenosis were measured. Each dot represents one animal ($n = 9$). **$P < 0.01$. Two-tailed unpaired $t$-test was used. **i** Time lapse imaging of thrombus formation in mesenteric arteries of Mfsd2bf/f (f/f) and Mfsd2bf/fPF4 (PF4-Cre) mice. **j** Occlusion time of blood flow in mesenteric artery during thrombosis from Mfsd2bf/f and Mfsd2bf/fPF4 mice. Each dot represents one mouse ($n = 6$ for Mfsd2bf/f; $n = 5$ for Mfsd2bf/fPF4). ***$P < 0.001$. Two-tailed unpaired $t$-test was used. **k** Thrombus sizes quantified by fluorescence intensity of the thrombus occurred at 5 min from Mfsd2bf/f and Mfs2bf/fPF4 miceEach dot represents one animal ($n = 6$). *$P < 0.05$. Two-tailed unpaired $t$-test was used. Scale bars in **i**: 100 μm. Data are mean and SD in all panels.

Furthermore, we generated and examined arterial and venous thrombosis in platelet-specific knockout of Mfsd2b (Mfsd2bf/fPF4). We found that Mfsd2bf/fPF4 mice have also significantly reduced venous thrombotic phenotypes compared with controls (Fig. 8h). In arterial thrombotic models, live imaging of thrombosis in mesenteric arteries in Mfsd2bf/f and Mfsd2bf/fPF4 mice showed that Mfsd2b deficient platelets had significantly reduced thrombotic functions (Fig. 8i–k). We observed that Mfsd2bf/fPF4 mice took more time to form thrombus (Fig. 8i, j). The size of thrombi from Mfsd2bf/fPF4 mice was significantly reduced compared with that of control mice (Fig. 8k). Similar results were also observed in a model for carotid arterial thrombosis (Supplementary Fig. 9) Together, our results indicate that loss of Mfsd2b causes defects in the intrinsic functions of platelets, which suppresses their thrombotic properties.

## Discussion

Sphingosine-1-phosphate plays pleiotropic signaling roles by activating five different receptors (S1P1-5). The roles of S1P signaling in vasculature and immune cell trafficking is well established by the characterizations of S1P receptor and sphingosine kinase (SphK1 and SphK2) knockout models. The recent identifications of S1P transporters further advance our understanding of the physiological roles of S1P signaling[5,6]. Circulating S1P is generated from various sources and blood cells are reported to contribute >50% of it[5]. In addition, platelets are also known to secrete S1P to blood. Studies in ex vivo conditions have indicated that platelets can store S1P and release this pool of S1P after activation[16]. In addition, S1P from platelets may have signaling roles. In this study, utilizing knockout mice for S1P transporter Mfsd2b and lipidomics, we show that Mfsd2b is an essential exporter of S1P in resting and activated platelets. However, our data show that platelets contribute a minimal amount of plasma S1P. Furthermore, our results indicate that circulating S1P from platelets is unlikely required for platelet signaling during thrombosis in DVT models. Instead, it might contribute to specific roles during systemic coagulation that occurs in sepsis or anaphylactic shock as reported previously[16]. We uncover unexpected effects from the Mfsd2b-S1P axis for platelet homeostasis. In summary, this work clarifies that (1) Mfsd2b is expressed in the plasma membrane in resting and activated platelets and that resting platelets store S1P the cytoplasm; (2) platelets are not the major provider of the plasma S1P pool and this pool is unlikely required for induction of venous thrombosis; (3) deletion of Mfsd2b causes S1P accumulation, which may trigger lipotoxic stress in platelets and severely impede their intrinsic thrombotic functions.

Platelets synthesize and release S1P to exert specific functions such as under anaphylactic shock conditions[16]. Mounting evidence have suggested that they export S1P in a protein-dependent fashion[10,21]. The identity of S1P transporter in platelets was unclear as a recent study showed that MRP4, an ABC transporter, is also involved in S1P release from platelets[17]. However, the inability to release S1P in both resting and activated platelets from Mfsd2b KO mice argues against the presence of other transporters, such as MRP4, in these cells[17]. Our work here shows that Mfsd2b is an essential transporter for S1P in platelets[5]. Furthermore, we show that resting and activated platelets release endogenous S1P species in an Mfsd2b-dependent manner. The accumulation of a substantial amount of S1P in Mfsd2b knockout platelets possibly highlights that Mfsd2b is a critical transporter for S1P in platelets and that S1P hydrolyzing enzymes are not expressed in platelets. Using Mfsd2b knockout platelets as a control, we also demonstrate that resting platelets

can store a detectable amount of S1P in the cytoplasm and release this pool upon activation. Upon activation, S1P is delivered via exocytosis to the plasma membrane, where Mfsd2b is localized. In support of this model, platelets deficient for SNARE components have a reduced S1P release due to impaired exocytosis[30,31]. Thus, it is likely that platelets can control the release of stored S1P pool via exocytosis. Although it remains to be characterized how S1P is generated in the intracellular compartments, we propose that exogenous sphingosines in endocytic vesicles are phosphorylated by SphK2, which is specifically expressed in platelets[9]. This conversion would generate S1P in the outer leaflet of endocytic vesicles. Exocytosis of these vesicles delivers S1P to the inner leaflet of the plasma membrane, where it can be exported to the outer leaflet of the plasma membrane by Mfsd2b. Our current results support the essential roles of Mfsd2b in this process.

Sphingosine-1-phosphate production has been linked with a role for platelets functions[16,32]. We also show that Mfsd2b deletion in platelets causes reduced aggregation. However, our data point to a notion that intrinsic defects in S1P export due to Mfsd2b deletion in platelets may explain for the predominant phenotype of reduced aggregation in Mfsd2b knockout platelets. S1P synthesis in megakaryocytes was shown to play a role in platelets biogenesis in global deletion of SphK2 knockout mice[32]. In contrast, specific deletion of SphK2 in platelets did not affect platelet biogenesis[15]. We do not observe any significant reduction in number of platelets at the steady state or the recovery state after induction of platelet depletion from global knockout or platelet-specific knockout of Mfsd2b. Thus, our observations suggest that Mfsd2b-S1P pathway is not required for platelet biogenesis, consistent with a recent report[15]. Why is there a thrombotic phenotype in Mfsd2b knockout platelets? Abrogation of S1P production in platelets by deleting SphK2 does not result in morphological changes of platelets[9]. In stark contrast, we found that Mfsd2b knockout platelets had morphological abnormalities. Additionally, specific deletion of SphK2 in platelets did not affect thrombotic functions[16], whereas loss of Mfsd2b in platelets results in significantly reduced thrombosis. The differences in thrombotic phenotypes between SphK2 and Mfsd2b knockout mice could be likely due to the accumulation of sphingosines and ceramides, and especially S1P species in Mfsd2b knockout platelets. We propose that the overload of these sphingolipids causes detrimental effects on platelet membranes, where adhesion and signaling molecules are localized. The lipotoxicity effects of sphingolipids in platelets might be similar to their effects observed for other cells[33,34]. In line with the defects induced by sphingolipid accumulation in platelets due to Mfsd2b loss, we also observed morphological defects in erythrocytes of Mfsd2b knockout mice[5]. S1P accumulation in the inner leaflet of platelets might have resulted in disorganizations of platelet membrane structures, which impedes their thrombotic functions.

Intriguingly, we show that whole-body Mfsd2b KO mice and Mfsd2b-specific deletion in platelets exhibit reduced venous thrombosis, whereas Mfsd2b expression in erythrocytes is dispensable for this process. We found that the total plasma S1P levels remains unchanged in thrombotic conditions. In line with this, thrombotic functions of platelets from erythrocyte-specific Mfsd2b knockout mice, which has a significantly reduced systemic S1P level, are not impaired. These results implicate that systemic S1P is unlikely required for thrombosis. Furthermore, although thrombosis is linked to the increased inflammation in vascular cells[35], we found that inhibition of S1P2, which is the S1P receptor strongly linked to vascular inflammation[28], does not reduce DVT. Rather, our results suggest that the reduced thrombosis in Mfsd2b knockout mice could be explained by

reduced adhesion of Mfsd2b KO platelets. The adhesion of platelets to thrombotic sites, mediated by binding of the platelet receptor Clec-2 to vascular derived podoplanin, is involved in the establishment of thrombosis[26,36]. We found that podoplanin expression is induced under thrombotic conditions in WT mice, but its expression is significantly reduced in KO mice. Furthermore, Mfsd2b knockout platelets have reduced aggregation with neutrophils after activation. Although our current results do not pinpoint the molecular mechanisms by which deletion of Mfsd2b in platelets results in impaired thrombosis, we propose that defects in the plasma membrane due to sphingolipid accumulation could affect platelets adhesion to each other or other vascular cells to form a thrombus. Nonetheless, since Mfsd2b inhibition also abrogates the release of S1P, we cannot rule out the local signaling effects of S1P released from platelets during thrombosis. In addition, it is unclear if Mfsd2b also mediates the interaction of platelets with vascular cells in thrombotic events. These warrant further investigations.

In summary, we report here the essential roles of Mfsd2b as an exporter for all S1P species found in platelets. Resting platelets store S1P in the cytoplasm and Mfsd2b is predominantly expressed in the plasma membrane. Deletion of Mfsd2b in platelets results in significantly impaired platelet functions. Inhibition of Mfsd2b in platelets does not affect systemic S1P levels, implying that targeting this protein for treatment of thrombosis may not affect signaling roles of circulating S1P. Our results show that targeting Mfsd2b functions in platelets may be explored for reducing thrombotic conditions.

## Methods

**Mice**. Knockout mice of Mfsd2b were generated as described, previously[5]. All mice were in C57BL/6 background. Platelet-specific knockout Mfsd2b mice were generated by inter-crossing homozygous floxed Mfsd2b (Mfsd2bf/f) mice with Mfsd2bf/f mice containing a copy of PF4-cre[37]. Mice carrying PF4-cre was genotyped by forward primer CCCATACAGCACACCTTTTG and reverse primer TGCACAGTCAGCAGGTT. Red blood cells (RBC) specific knockout of Mfsd2b with EpoR-GFP-cre mice was genotyped by forward primer GTGTGGCTGCC CCTTCTG and reverse primer CAGGAATTCAAGCTCAACCTCA[38]. Deletion of Mfsd2b in RBC and platelets was confirmed by western blot and S1P transport activity. Mice were maintained in normal chow diets. All experiments performed in this study were approved in the breeding protocol BR19-0633 and research protocol R19-0567 by the Institutional Animal Care and Use Committee (IACUC) of the National University of Singapore.

**Deep vein thrombosis**. The flow restriction (stenosis) model of DVT was performed following Holly Payne's protocol with minor modification[24]. Briefly, mice were anesthetized using isoflurane and kept in a supine position with a mask attached to give a constant supply of anesthesia. The inferior vena cava (IVC) was exposed and gently separated from aorta and a suture was made over a 30-gauge needle to achieve 90% closure. After 48 h, the mice were killed and thrombi in the IVC were removed, weighed, and stained by hematoxylin and eosin. Experiments were blinded.

**Statistical analysis**. Data were analyzed using GraphPrism8 software for Windows. Statistical significance was calculated using two-tailed unpaired $t$-test or one and two-way ANOVA as indicated in the figure legends. $P$ value $< 0.05$ was considered as statistical significance. Detailed method sections can be found in Supplementary information.

**Reporting summary**. Further information on research design is available in the Nature Research Reporting Summary linked to this article.

## Data availability

All data are provided in the manuscript and the raw data have been provided in the source data file. Source data are provided with this paper.

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

## Acknowledgements

We thank Dr. Deron Herr for S1P2 knockout mice and wild-type controls. We thank Prof. Shigekazu Nagata for Tmem16F antibody. This study was supported in part by Singapore Ministry of Health's National Research Council NMRC/OFIRG/0066/20, Ministry of Education MOE2018-T2-1-126, MOE-Tier-1, MOE PDF fellowship, NUS Young Investigator Award (NUSYIA_FY16_P19), and NUHSRO/2017/066/BRIDGING/04, grants (to L.N.N.), Ministry of Education MOE-Tier-1 (R-181-000-183-114) (to W.Y.O.). We thank Ms. Y. J. Wu (Anatomy, NUS) for technical assistance with SEM and TEM.

## Author contributions

M.C. performed in vivo experiments and T.Q.N. and T.M.V. performed ex vivo experiments; M.C., T.M.V, and T.Q.N. prepared samples for lipidomics; S.H.T. and A.W.L.L assisted with aggregation assays. S.B., Z.H., H.T.T.H., and U.T.N.L. performed in vivo and ex vivo experiments. F.T., A.C.-G., S.M., and J.C.F performed lipidomic analysis and compiled data. Z.H. and T.Q.N performed mesenteric arterial thrombosis. M.C. and Z.H. performed flow cytometry experiments. M.R.W. supervised F.T. and A.C.-G; W.Y.O. analyzed TEM and SEM images; M.Y.Y.C. supervised S.H.T. and A.W.L.L.; L.N.N. conceived and designed the study and experiments; analyzed all data and wrote the paper.

## Competing interests

The authors declare no competing interests.
