## [Peer Review File · Nature Communications]

Reviewers' comments:

Reviewer #1 (Remarks to the Author):

This manuscript from the group that originally described the second true S1P transporter shows that Mfsd2b function is essential for S1P export in platelets and this mechanism is important for platelet functions during thrombosis. They attempt to address the molecular mechanism, which seems to involve an accumulation of sphingolipids, membrane structural alterations and reduced platelet membrane proteins (2b3a, P-selectin) and suggest that sphingolipid-induced platelet lipotoxicity may be the main culprit.

This is an important study that revealed a novel function of Mfsd2b in platelets and thrombosis. Overall, the data support the main conclusions and the quality of the work is high.

There are some issues that weaken some of the conclusions nevertheless, and I suggest that the authors address these.

1. Based on NBD-sphingosine uptake and optical microscopy, the authors suggest that S1P is stored in granules and that upon activation stored S1P somehow reaches to the Mfsd2b in the plasma membrane. This is not supported by data. First, NBD-sphingosine is not equivalent to sphingosine. Second, the localization studies of Mfsd2b are weak. The cell surface biotinylation data are equivocal and accurate quantitation is not provided. Immunofluorescence studies of the transporter may yield more definitive data. Alternatively purified granule fraction from resting and activated platelets can be analyzed by markers and Mfsd2b.

2. Figures - the authors should show absolute amounts of sphingolipids quantified by LC/MS

3. radioactive sphingosine experiments should show total label incorporated per million platelets in WT and KO.

4. It is recommended that the authors consider either deleting figure 2b or at least moving it to the supplement section. Conclusions should be toned down/ deleted.

5. The extent of genetic deletion of the Mfsd2b locus in Pf4-Cre and Epor-Cre mice in the appropriate cell populations should be shown.

6. better discussion (more detailed or accurate) is needed in the following places:

- line 195 - discuss potential molecular mechanism
- line 206- better description of "damaged membrane"
- line 254-256 - better description of DVT/ PDPN results needed
- lines 297-299 - further discussion needed/ recommended
- line 304- please edit

Other issues

1. More lucid discussion of other purported transporters vs. Mfsd2b in platelets needed.

2. The entire text should be checked for grammar, etc. and thoroughly edited.

Reviewer #2 (Remarks to the Author):

The present manuscript authored by Nguyen and coworkers describe that the sphingosine-1-phosphate (S1P) exporter Mfsd2b regulates sphingolipids in platelets which is associated with functional defects of platelet function. Genetic deletion of Mfsd2b in platelets reduces platelet aggregation and venous thrombosis in mice without affecting systemic plasma S1P levels. The authors claim Mfsd2b as a novel target to modulate thrombosis.

Previously, the research group showed that Mfsd2b is a critical regulator of S1P secretion of blood cells (erythrocytes and platelets) which plays a role in red blood cell morphology and hemolysis (Nature 2017). The present study describes follow-up data and concentrates on the role of Mfsd2b and S1P on platelet function. As already shown for red blood cells the authors now provide data that similar to erythrocytes deletion of Mfsd2b leads to accumulation of S1P in platelets. In contrast to their findings with erythrocytes, deletion of Mfsd2b in platelets does not result in enhanced release of S1P from activated platelets.

General comments

The results are interesting, however, the observed impairment of platelet function and thrombus formation is not well characterized. It is not clear how deficiency of Mfsd2b regulates aspects of platelet function. At this state the data remains descriptive. The authors should elaborate a clear molecular mechanism how Mfsd2b/S1P regulates platelet function.

Specific comments

1. The authors speculate that Mfsd2b is associate with the platelet plasma membrane, however, the data shown in Figure 2 are not totally convincing. Subcellular fractionation of platelet organelles and the plasma membrane following by immunoblotting might be more appropriate to approach this issue. Do the authors have data that provide evidence that Mfsd2b is exclusively found in the plasma membrane or is also present in granula membranes (e.g. alpha-granules or delta-granules)? This may be important to provide a mechanism how S1P is stored in platelets? Further the super-resolution imaging is not totally convincing. Using NBD-sphingosine as substrate for S1P synthesis (Figure 2B) shows that fluorescence signals are found in some granula-like structure but also in the cytoplasm. Almost no fluorescence signals are visible in the plasma membrane. It would be of interest to see potential changes in fluorescence signal patterns in resting and activated platelets derived from wildtype and Mfsd2b-KO mice. Further co-staining with granula markers might help to define the nature of the granula-like structures.
2. Figure 3. The authors show that in Mfsd2b-KO mice S1P species are retained in the platelet compartment upon activation. It would be of interest whether this is primarily due to the absence of the Mfsd2b transporter or whether the Mfsd2b-KO platelets have a defect in the release reaction of granula. Since the morphology of the Mfsd2b-KO platelets is significantly altered a granula-release defect cannot be excluded. Data showing the surface expression of e.g. P-selectin and release of ATP and serotonin of resting and activated platelets might be of interest to address this issue.
3. Figure 4A shows that thrombin-activated platelets deficient in Mfsd2b exhibit a reduced release of S1P into the supernatant. It would be interesting to see whether this is a result of a deficiency of active transport activity. The authors need to show that the release of granula is unaltered which could be significantly contribute to reduction in S1P release (see Figure 5C). In this case reduction of S1P upon platelet activation might not be caused by altered Mfsd2b but just be a secretion defect.
4. Figure 5 shows that aggregation of whole blood is reduced in Mfsd2b-KO mice. It is not clear whether the authors used the global or the platelet-specific Mfsd2b-deletion mouse strain. To assess whether Mfsd2b regulates platelet function further functional data e.g. platelet-dependent thrombus formation under flow and also in vivo data on thrombus formation needs to be explored to give a more detailed information on the impairment of platelet function. Further, it would be of interest whether the observed functional platelet defect can be rescued by supplementation of S1P species?
5. Figure 8. It is not totally clear how the platelet transfusion experiments were performed? How many platelets were transferred into the platelet depleted mice? Did the authors check whether transfusion of platelets resulted in a significant increase in circulating platelets? At what time after initial administration of the depleting CD42b antibody did the authors start with the platelet transfusion? This is critical since the CD42b antibody has a significant plasma half time that might also affect the transfused platelets.

Reviewers' comments:

Reviewer #1 (Remarks to the Author):

This manuscript from the group that originally described the second true S1P transporter shows that Mfsd2b function is essential for S1P export in platelets and this mechanism is important for platelet functions during thrombosis. They attempt to address the molecular mechanism, which seems to involve an accumulation of sphingolipids, membrane structural alterations and reduced platelet membrane proteins (2b3a, P-selectin) and suggest that sphingolipid-induced platelet lipotoxicity may be the main culprit.

This is an important study that revealed a novel function of Mfsd2b in platelets and thrombosis. Overall, the data support the main conclusions and the quality of the work is high.

There are some issues that weaken some of the conclusions nevertheless, and I suggest that the authors address these.

1. Based on NBD-sphingosine uptake and optical microscopy, the authors suggest that S1P is stored in granules and that upon activation stored S1P somehow reaches to the Mfsd2b in the plasma membrane. This is not supported by data. First, NBD-sphingosine is not equivalent to sphingosine. Second, the localization studies of Mfsd2b are weak. The cell surface biotinylation data are equivocal and accurate quantitation is not provided. Immunofluorescence studies of the transporter may yield more definitive data. Alternatively purified granule fraction from resting and activated platelets can be analyzed by markers and Mfsd2b.

Response: *We thank the reviewer for the comment. For the immunofluorescence studies, we have generated a few different antibodies for Mfsd2b. Unfortunately, none of the antibodies works for immunostaining with platelets. They only works with protein samples for western blot (e.g. Fig 1A). To answer your comments, we performed biontynylation experiments and subcellular fractionation experiments as suggested. Our new data from both experiments indeed support the conclusion that Mfsd2b is predominantly localized in the plasma membrane of platelets (New Fig 2). There is a detectable amount of Mfsd2b found in the unbound and cytosolic fractions from these experiments. This could be due to the incomplete recovery of Mfsd2b in the plasma membrane in these experiments or indeed a small amount of Mfsd2b is also present in the cytoplasm of platelets. Nevertheless, after normalization expression levels of Mfsd2b in the plasma membrane of resting platelets is greater than its intracellular levels.*

With regarding to S1P localization, we performed new experiments in which resting platelets were loaded with radioactive S1P. The cytosolic and membrane fractions were harvested via freeze-thaw method. We found that radioactive S1P levels in cytosolic fractions are significantly higher than that of membrane fractions (Fig 2H). We also measured endogenous S1P from cytosolic and membrane fractions by mass spectrometry. S1P is present with higher levels in cytosolic fractions of resting WT and Mfsd2b KO platelets (Fig 2I, J). Our data show that in resting platelets Mfsd2b and S1P are predominantly present in the plasma membrane and cytoplasm, respectively. These data are added to the manuscript. For the utilization of NBD-sph, we agree to remove this data from manuscript.

2. Figures - the authors should show absolute amounts of sphingolipids quantified by LC/MS

Response: *We thank the reviewer for the comment. We have revised the figures as requested.*

3. radioactive sphingosine experiments should show total label incorporated per million platelets in WT and KO.

Response: *We would like to explain the presentation of data for these experiments. In the time course experiments (Fig 1D, E, also new data), we activated before using them for our transport assays. In this case, we can examine whether activated WT platelets can store S1P or not. The radioactive signals were presented in DPM/million platelets. We would like to show the radioactive signals from supernatant and cell pellet separately. This is to show that there is a defect in release of S1P in KO platelets. The actual radioactive counts are used in the figure.*

4. It is recommended that the authors consider either deleting figure 2b or at least moving it to the supplement section. Conclusions should be toned down/ deleted.

Response: *We thank the reviewer for the comment. We agree to remove Figure 2b from our manuscript as we have provided with new data to replace this data. We also revised our text as suggested accordingly.*

5. The extent of genetic deletion of the Mfsd2b locus in Pf4-Cre and Epor-Cre mice in the appropriate cell populations should be shown.

Response: *We thank the reviewer for the comment. We performed these experiments. The data are included in the manuscript (Please see Fig 4A, B).*

6. better discussion (more detailed or accurate) is needed in the following places:
-line 195 - discuss potential molecular mechanism.

Response: *We have discussed about the potential mechanism for the observed phenotypes. Please see in discussion line 376-379.*

- line 206- better description of "damaged membrane"

Response: *We have revised the text accordingly.*

- line 254-256 - better description of DVT/ PDPN results needed

Response: *We have revised the text accordingly.*

- lines 297-299 - further discussion needed/ recommended

Response: *We have revised the text accordingly.*

- line 304- please edit

Response: *We have revised the text accordingly.*

Other issues

1. More lucid discussion of other purported transporters vs. Mfsd2b in platelets needed.

Response: *We thank the reviewer for the suggestion. We have added this information to the discussion.*

2. The entire text should be checked for grammar, etc. and thoroughly edited.

Response: *We have revised the text accordingly.*

Reviewer #2 (Remarks to the Author):

The present manuscript authored by Nguyen and coworkers describe that the sphingosine-1-phosphate (S1P) exporter Mfsd2b regulates sphingolipids in platelets which is associated with functional defects of platelet function. Genetic deletion of Mfsd2b in platelets reduces platelet aggregation and venous thrombosis in mice without affecting systemic plasma S1P levels. The authors claim Mfsd2b as a novel target to modulate thrombosis.

Previously, the research group showed that Mfsd2b is a critical regulator of S1P secretion of blood cells (erythrocytes and platelets) which plays a role in red blood cell morphology and hemolysis (Nature 2017). The present study describes follow-up data and concentrates on the role of Mfsd2b and S1P on platelet function. As already shown for red blood cells the authors now provide data that similar to erythrocytes deletion of Mfsd2b leads to accumulation of S1P in platelets. In contrast to their findings with erythrocytes, deletion of Mfsd2b in platelets does not result in enhanced release of S1P from activated platelets.

General comments

The results are interesting, however, the observed impairment of platelet function and thrombus formation is not well characterized. It is not clear how deficiency of Mfsd2b regulates aspects of platelet function. At this state the data remains descriptive. The authors should elaborate a clear molecular mechanism how Mfsd2b/S1P regulates platelet function.

Specific comments

1. The authors speculate that Mfsd2b is associate with the platelet plasma membrane, however, the data shown in Figure 2 are not totally convincing. Subcellular fractionation of platelet organelles and the plasma membrane following by immunoblotting might be more appropriate to approach this issue.

Response: *We thank the reviewer for the comment. We performed additional biotinylation experiments as well as performed new subcellular fractionation experiments. We showed that Mfsd2b is predominantly localized in the plasma membrane in resting and activated platelets (New Fig 2). There was a small amount of Mfsd2b in the cytosolic and unbound fraction. This could be due to the incomplete recovery of Mfsd2b in the assays or indeed a small amount of Mfsd2b is present in the cytoplasm. Nevertheless, expression of Mfsd2b is significantly higher in the plasma membrane. As Mfsd2b is present in the plasma membrane of resting platelets, we argue that if Mfsd2b is present in the plasma membrane and if S1P is also present in the same location, S1P would be transported in resting platelets. However, a large amount of S1P is stored in resting platelets. To support for this argument, we obtained*

new data to show that S1P is mainly stored in cytoplasm in resting platelets (Fig 2H-J). We have added the new data and revised the text accordingly.

Do the authors have data that provide evidence that Mfsd2b is exclusively found in the plasma membrane or is also present in granula membranes (e.g. alpha-granules or delta-granules)? This may be important to provide a mechanism how S1P is stored in platelets?

Response: *We thank the reviewer for the insightful comment. As our data suggest Mfsd2b is mainly expressed in the plasma membrane (Fig 2). It is difficult to obtain clean data to show that Mfsd2b is exclusively expressed in the plasma membrane, especially with platelets. Our biotinylation data show that a detectable amount of Mfsd2b is present in the cytosolic fractions. Thus, we cannot rule out if Mfsd2b is also present inside the cells. Based on new data in Fig 2, we conclude that expression levels of Mfsd2b are significantly higher in the plasma membrane compared with its intracellular levels in resting platelets, whereas S1P is mainly present in the cytosolic fractions in resting platelets. This would explain for the low S1P transport activity in resting platelets. We agree with the reviewer that we cannot say that Mfsd2b is exclusively expressed in the plasma membrane. We revised the text accordingly.*

Further the super-resolution imaging is not totally convincing. Using NBD-sphingosine as substrate for S1P synthesis (Figure 2B) shows that fluorescence signals are found in some granula-like structure but also in the cytoplasm. Almost no fluorescence signals are visible in the plasma membrane. It would be of interest to see potential changes in fluorescence signal patterns in resting and activated platelets derived from wildtype and Mfsd2b-KO mice. Further co-staining with granula markers might help to define the nature of the granula-like structures.

Response: *We agreed with the reviewer's comment about this experiment. Due to technical limitations, platelets are too small to obtain convincing immunostaining data. Our polyclonal antibodies don't work with immunofluorescent staining for platelets. We agree that NBD-sphingosine might behave differently. Reviewer 1 also raised the same comment that we agreed. We removed this data from the manuscript as suggested by reviewer 2.*

Instead, we performed subcellular fractionation experiments of endogenous S1P from cytosolic and membrane fractions and quantified by mass spectrometry (Fig 2I,J). We showed that in resting platelets S1P levels in the cytosolic fractions are significantly higher than that in the membrane fractions. We also performed similar experiments with radioactive S1P. In these experiments, [3H]-S1P from cytosolic and membrane fractions from resting platelets were harvested. Similar results were also obtained (Fig 2H). In summary, in resting platelets S1P levels in cytosolic fraction is significantly higher than in the membrane fractions. These data are added to the current manuscript.

2. Figure 3. The authors show that in Mfsd2b-KO mice S1P species are retained in the platelet compartment upon activation. It would be of interest whether this is primarily due to the absence of the Mfsd2b transporter or whether the Mfsd2b-KO platelets have a defect in the release reaction of granula. Since the morphology of the Mfsd2b-KO platelets is significantly altered a granula-release defect cannot be excluded. Data showing the surface expression of e.g. P-selectin and release of ATP and serotonin of resting and activated platelets might be of interest to address this issue.

Response: We thank the reviewer for the constructive comment. In the activation conditions for Figure 3, thrombin was used to activate platelets. Thrombin-activated Mfsd2b KO platelets released PF4 that can be detected by Western blot. As suggested by the reviewer, we performed experiments to examine the release of ATP, serotonin, and PF4 (Supplemental fig 2). We show that in thrombin-activated conditions, the release of ATP, serotonin, and PF4 in KO platelets was comparable to that of WT platelets. These data indicate that the release of the soluble compounds contained in the granula in KO platelets was not reduced under the activation conditions. In contrast, our data show that S1P which is a hydrophobic molecule associated with the membranous compartments was not released in Mfsd2b KO platelets under these conditions. For P-selectin surface expression, we found that reduced surface expression of this molecule under calcium ionophore activation (Fig 5D). We used this activation condition to examine whether there is a reduction of P-selectin in KO platelets compared with WT platelets. We also noted that activation of platelets with calcium ionophore did not result in a complete release of S1P compared with thrombin (New Fig 4C and Vu et al., Nature, Figure 3g). Therefore, we used thrombin-activated platelets in the experiments for transport assays as well as for lipidomics analysis in Figure 3.

3. Figure 4A shows that thrombin-activated platelets deficient in Mfsd2b exhibit a reduced release of S1P into the supernatant. It would be interesting to see whether this is a result of a deficiency of active transport activity. The authors need to show that the release of granula is unaltered which could significantly contribute to reduction in S1P release (see Figure 5C). In this case reduction of S1P upon platelet activation might not be caused by altered Mfsd2b but just be a secretion defect.

Response: We thank the reviewer for the comment. As we have answered in the question 2 above that when we used thrombin for platelet activation, we observed that PF4 release is not affected in KO platelets. We also showed ATP and serotonin were strongly released with thrombin activation in Mfsd2b KO platelets. However, Mfsd2b KO platelets failed to release S1P with thrombin activation conditions. Thus, these data support the conclusion that Mfsd2b KO platelets are unable to release S1P, but they can still release granular content in thrombin activation conditions.

4. Figure 5 shows that aggregation of whole blood is reduced in Mfsd2b-KO mice. It is not clear whether the authors used the global or the platelet-specific Mfsd2b-deletion mouse strain. To assess whether Mfsd2b regulates platelet function further functional data e.g. platelet-dependent thrombus formation under flow and also in vivo data on thrombus formation needs to be explored to give a more detailed information on the impairment of platelet function.

Response: We thank the reviewer for the suggestive comment. The results from Figure 5 were from the whole body knockout of Mfsd2b mice that we used for most experiments in the current manuscript.

Regarding to the in vivo thrombosis formation experiments, we performed these experiments as suggested. These include carotid artery thrombosis (Supplemental fig 9) and intravital microscope imaging of thrombus formation using FeCl₃ injury method in mesenteric arteries of Mfsd2b/f control and Mfsd2b/f/PP4 mice. We found that thrombus formation in Mfsd2b/f/PP4 mice was significantly delayed compared with controls (Fig 8I-K). These additional data strengthen our conclusions that deletion of Mfsd2b affects thrombotic functions of platelets.

Further, it would be of interest whether the observed functional platelet defect can be rescued by supplementation of S1P species?

Response: *We thank the reviewer for the insightful comment. This was also our initial hypothesis that inhibition of S1P release from platelets would affect S1P signalling in platelets. However, we failed to obtain rescue effects from addition of 2uM S1P to the reduced aggregation phenotypes that we have observed from the whole blood of global Mfsd2b KO (where plasma S1P is reduced by 50%). There are also evidences from our data suggesting that reduced extracellular S1P levels might not be the cause for defective platelet functions in Mfsd2b knockout mice: 1) Platelets from KO mice have defects in morphology. This phenotype is not present in SphK2 knockout platelets, 2) Reduction of plasma S1P in Mfsd2b^{fl/fl} EpoR-cre mice did not affect thrombosis in DVT model, 3) Plasma S1P was not reduced in WT mice underwent DVT even though number of circulating platelets was reduced. We believe that the remaining S1P amount in plasma of Mfsd2b deficient mice would saturate S1P receptors for signaling. Thus, we pursued that the notion that intrinsic defects in Mfsd2b KO platelets are linked to its reduced functions.*

5. Figure 8. It is not totally clear how the platelet transfusion experiments were performed? How many platelets were transferred into the platelet depleted mice? Did the authors check whether transfusion of platelets resulted in a significant increase in circulating platelets? At what time after initial administration of the depleting CD42b antibody did the authors start with the platelet transfusion? This is critical since the CD42b antibody has a significant plasma half time that might also affect the transfused platelets.

Response: *We thank the reviewer for the comment. We apologize for the confusions. We used CD42b to deplete platelets for platelet recovery experiments. The data were not included in the manuscript (see below). We noted that CD42b treatment causes mild symptoms in our platelet recovery experiments. The half-life of CD42b might deplete transfused platelets as well. Thus, we injected 800 million of platelets per mouse without platelet depletion and performed DVT in the same day. As the blood drawing would interfere with the thrombosis in the mice, we did not examine the number of platelets after platelet transfusion. We believe that injection of 800million of platelets will increase amount of circulating platelets in the recipient mice in the DVT experiments.*

Figure for reviewing purpose. Deletion of *Mfsd2b* did not affect platelet recovery from megakaryocytes. *A*, *Mfsd2b* had a normal platelet recovery from an acute depletion of platelets with CD42 antibody. $n = 5-7$ mice per each genotype. *B*, Number of megakaryocytes from bone marrow of KO mice from normal condition or after an acute platelet depletion was comparable with WT mice. CD41 was used to stain for megakaryocytes. Laminin1 and Ki67 were used to stain for blood vessels and stem cells, respectively. Representative images from 2 WT and 2 KO for each conditions.

REVIEWER COMMENTS

Reviewer #1 (Remarks to the Author):

The revisions are adequate and fully address my previous concerns.

Reviewer #2 (Remarks to the Author):

All my comments have been adequately addressed. I don't have further suggestions.

Reviewer #3 (Remarks to the Author):

The Manuscript of Chandrakanthan et al presents a series of analyses that link the deletion of Mfsd2b, S1P and the thrombotic function of platelets. The manuscript is well written and clearly describes all of the experiments. The methodology description is less complete and I focus here on the lipidomic analyses. The methods are brief and reference previous papers 2014 and 2016. Unless the exact same methods were used, any changes to the methods should be included in this manuscript. Further to this, the exact species that were measured and summed to calculate the lipid classes must be reported somewhere in this manuscript. I could not see these, even in the earlier references.

Further on the lipidomics analyses, the units given are nM/million platelets. This does not make sense; you cannot report a concentration per million platelets. This should be nmol/million platelets I believe. Please correct this.

I think there may be an opportunity missed here as the authors have measured multiple lipid species but not shown the analyses of the individual lipid species. Were there any changes in the composition of the species within the PC or PE classes for example. Given the earlier report of Mfsd2a being a transporter for DHA, I wonder if the KO would have some different lipidomic profile in terms of the fatty acid composition of the membrane lipids rather than just looking at the lipid class level. This could easily be checked and if negative could be noted here.

This brings me to my major concern over this manuscript and that is the lack of mechanistic link between the effect of Mfsd2b on S1P, which I think is well demonstrated, and the effect on thrombosis which is also well demonstrated. What is lacking in any real evidence that it is the build-up of S1P within the platelets that is driving this. The idea that S1P is impacting membrane integrity may be true but it is likely one of several possible explanations of why the Mfsd2b KO have impaired thrombosis activity, particularly given the role of Mfsd2b as a mediator of exocytosis. It would have been good to see that, for example, if the synthesis of S1P was inhibited then the effect Mfsd2b KO on thrombosis was abrogated. This would provide the necessary link between S1P and thrombosis.

By the same argument the effect of S1P on the cell morphology is also questionable.

Thus as a minimum, I think this manuscript requires a more complete discussion about the possible roles of Mfsd2b in platelets to better balance the arguments for S1P as a driver of the cell morphology and thrombosis effects. Better still would be additional experiments to more tightly link these effects.

REVIEWER COMMENTS

Reviewer #1 (Remarks to the Author):

The revisions are adequate and fully address my previous concerns.

Reviewer #2 (Remarks to the Author):

All my comments have been adequately addressed. I don't have further suggestions.

Reviewer #3 (Remarks to the Author):

The Manuscript of Chandrakanthan et al presents a series of analyses that link the deletion of Mfsd2b, S1P and the thrombotic function of platelets. The manuscript is well written and clearly describes all of the experiments. The methodology description is less complete and I focus here on the lipidomic analyses. The methods are brief and reference previous papers 2014 and 2016. Unless the exact same methods were used, any changes to the methods should be included in this manuscript. Further to this, the exact species that were measured and summed to calculate the lipid classes must be reported somewhere in this manuscript. I could not see these, even in the earlier references.

Response: *We thank the reviewer for the comment. We have revised the methods for lipidomics to provide with detail information. Regarding to the exact species, we apologize for the unclear presentation of the data in the previous submission. In this revision, we provide the raw data for these lipidomic analysis. The levels of individual lipid species are provided in the excel sheets along with normalized data used to generate the graphs.*

Further on the lipidomics analyses, the units given are nM/million platelets. This does not make sense; you cannot report a concentration per million platelets. This should be nmol/million platelets I believe. Please correct this.

Response: *We thank the reviewer for the comment. We have corrected the units for the levels of S1P from platelets.*

I think there may be an opportunity missed here as the authors have measured multiple lipid species but not shown the analyses of the individual lipid species. Were there any changes in the composition of the species within the PC or PE classes for example. Given the earlier report of Mfsd2a being a transporter for DHA, I wonder if the KO would have some different lipidomic profile in terms of the fatty acid composition of the membrane lipids rather than just looking at the lipid class level. This could easily be checked and if negative could be noted here.

Response: *We apologize for the unclear presentation of the data. As explained above, we have included the levels of individual lipid species in the excel sheets in the supplemental information. We also provided the raw data for these analyses.*

With regarding to the composition of phospholipids including PC, PE, PS, PI from Mfsd2b KO platelets, we performed these lipidomic analyses in order to gain insights if deletion of Mfsd2b also affects phospholipids. We have observed that there were a few phospholipid species from PC and PE that are slightly increased in the Mfsd2b KO platelets compared to WT platelets. For example, for PC 34:2 and PE18:0/22:6 species; these lipid species are slightly, but significantly increased in Mfsd2b KO

platelets. However, we speculate that these slight changes in phospholipid profile from *Mfsd2b* KO platelets are likely due to the increased volumes in *Mfsd2b* KO platelets (See supplemental Figure 5). In summary, we have not seen any major changes in phospholipid profile (e.g. PC, PE, PS, PI, LPC) from *Mfsd2b* KO platelets (and erythrocytes). Our results showed that only S1P, Sph, and Cer are accumulated in *Mfsd2b* KO platelets. These results highlight the role of *Mfsd2b* as a transporter for S1P, but not phospholipids such as LPC, which is transported by *Mfsd2a*.

This brings me to my major concern over this manuscript and that is the lack of mechanistic link between the effect of *Mfsd2b* on S1P, which I think is well demonstrated, and the effect on thrombosis which is also well demonstrated. What is lacking in any real evidence that it is the build-up of S1P within the platelets that is driving this. The idea that S1P is impacting membrane integrity may be true but it is likely one of several possible explanations of why the *Mfsd2b* KO have impaired thrombosis activity, particularly given the role of *Mfsd2b* as a mediator of exocytosis. It would have been good to see that, for example, if the synthesis of S1P was inhibited then the effect *Mfsd2b* KO on thrombosis was abrogated. This would provide the necessary link between S1P and thrombosis. By the same argument the effect of S1P on the cell morphology is also questionable.

Response: We thank the reviewer for the insightful comment. We agree with the reviewer's insight that in order to gain further evidence on the detrimental defects caused by S1P accumulation, inhibition of S1P synthesis in *Mfsd2b* KO platelets would normalize the observed phenotypes. The transport activity of *Mfsd2b* is relied on the source of S1P generated by *SphK2* in platelets. As such, deletion of *SphK2* in platelets would abrogate the accumulation of S1P regardless of the presence or absence of *Mfsd2b*. Previously published results on the roles of *SphK2*, the S1P synthesis enzyme in platelets, have demonstrated that lack of S1P synthesis via deletion of *SphK2* doesn't affect thrombotic functions of platelets. These works (PMID: 26129975 and PMID: 27582371) have shown that inhibition of S1P synthesis in platelets by deletion of *SphK2*, results in significantly low levels of S1P (100-fold reduction). However, unlike *Mfsd2b* KO platelets, *SphK2* KO platelets exhibit normal morphology (Figure 1 in PMID: 26129975). More importantly, lack of S1P synthesis in platelets also does not result in thrombotic phenotypes (Figure 5E, F in PMID: 27582371 and PMID: 31171507). Our data show that *SphK2* activity in *Mfsd2b* KO platelets is intact (e.g. *Mfsd2b* KO platelets can synthesize radioactive S1P from sphingosine (e.g. Figure 1D, Figure 4C)) that ensures to a normal S1P production. These give rise to the accumulation of S1P in *Mfsd2b* KO platelets as observed in our lipidomic analysis (Fig 3). We believe that these findings have provided strong evidence to support for our notion that accumulation of S1P is a major cause of observed phenotypes in *Mfsd2b* KO platelets. We have discussed these points in our revised manuscript.

With regarding to the role of *Mfsd2b* in exocytosis, our results show that *Mfsd2b* is not required for exocytosis because the release of ATP, serotonin, PF4 in *Mfsd2b* KO platelets are comparable to that of WT cells (Supplemental Fig 2). Our results show that *Mfsd2b* is predominantly expressed in the plasma membrane of both resting and activated platelets where it mediates the release of S1P. Thus, loss of *Mfsd2b* causes a block in S1P release via the plasma membrane. We only found that S1P and in some extents ceramides and sphingosines are significantly accumulated, but not for SM and phospholipids (PC, PE, PI, PS, PG, LPC). The degree of S1P accumulation in *Mfsd2b* KO platelets is remarkable (~10-fold and ~44-fold increase in resting and activated platelets, respectively). Thus, we believe that the accumulation of S1P species in *Mfsd2b* KO platelets is likely the major factor that causes membrane defects in *Mfsd2b* KO platelets. We also observed detrimental effects of S1P to erythrocytes (PMID: 29045386).

Thus as a minimum, I think this manuscript requires a more complete discussion about the possible roles of *Mfsd2b* in platelets to better balance the arguments for S1P as a driver of the cell

morphology and thrombosis effects. Better still would be additional experiments to more tightly link these effects.

Response: *We thank the reviewer for the comment. As we explained above that lack of S1P synthesis in platelets by deletion of SphK2 does not affect thrombotic functions of platelets. We only observed these thrombotic phenotypes in Mfsd2b KO platelets. Thus, we feel that performing additional experiments in which SphK2 is inhibited in Mfsd2b KO cells is not critical at this point. Instead, we have discussed these points in our revised manuscript.*

REVIEWERS' COMMENTS

Reviewer #3 (Remarks to the Author):

The authors have adequately addressed my concerns.

REVIEWERS' COMMENTS

Reviewer #3 (Remarks to the Author):

The authors have adequately addressed my concerns.